# Multi-modal sensor fusion towards three-dimensional airborne sonar imaging in hydrodynamic conditions

Aidan Fitzpatrick [1✉], Roshan P. Mathews [2], Ajay Singhvi [1] & Amin Arbabian [1]

Analogous to how aerial imagery of above-ground environments transformed our understanding of the earth's landscapes, remote underwater imaging systems could provide us with a dramatically expanded view of the ocean. However, maintaining high-fidelity imaging in the presence of ocean surface waves is a fundamental bottleneck in the real-world deployment of these airborne underwater imaging systems. In this work, we introduce a sensor fusion framework which couples multi-physics airborne sonar imaging with a water surface imager. Accurately mapping the water surface allows us to provide complementary multi-modal inputs to a custom image reconstruction algorithm, which counteracts the otherwise detrimental effects of a hydrodynamic water surface. Using this methodology, we experimentally demonstrate three-dimensional imaging of an underwater target in hydrodynamic conditions through a lab-based proof-of-concept, which marks an important milestone in the development of robust, remote underwater sensing systems.

---

[1] Department of Electrical Engineering, Stanford University, Stanford, CA, USA. [2] Department of Electrical Engineering, Indian Institute of Technology Palakkad, Kerala, India. ✉email: ajfitz@stanford.edu

Accelerated climate change has left humanity at a crucial inflection point in our history, with urgent calls for enhanced environmental monitoring and action[1–5]. Oceans play a critical role in our ecosystem—they regulate weather and global temperature, serve as the largest carbon sink and the greatest source of oxygen[6,7]. Despite that, greater than 80% of the ocean remains unobserved and unmapped today[8]. Thus, it is imperative that we develop means to reliably and frequently sense the rapidly changing ocean biosphere at a large-scale[9]. Remote sensing of the ocean ecosystem also has high-impact applications in various other spheres: disaster response, biological survey, archaeology, wreckage searching, among others[10–14].

Sonar is a mature technology that offers impressive high-resolution imaging of underwater environments[15,16]; however, its performance remains fundamentally constrained by the carrying vehicle. Typically, sonar systems are mounted to or towed by a ship that traverses an area of interest which limits frequent measurements and spatial coverage to a fraction of global waters[8]. A paradigm shift in how we sense underwater environments is needed to bridge this large technological gap. Radar[17], lidar[18], and photographic imaging systems[19] have enabled frequent, full-coverage measurements of the entire earth's landscapes, providing above-ground information on a global scale[20]. Likewise, there is a great push to develop remote underwater imaging systems which could have a similar transformative effect in imaging and mapping underwater environments.

Today, airborne lidar is the primary imaging modality used for imaging underwater from aerial systems[21,22]. These lidar systems exploit blue-green lasers which in clear waters are capable of penetrating as deep as 50 m[23,24]. Unfortunately, most water is not clear, particularly coastal waters, which have high levels of turbidity and can restrict the light penetration to less than 1 m[25,26], making lidars unsuitable for use in a large proportion of underwater environments. To exploit the advantages of in-water sonar, while operating aerially, some researchers have explored approaches that use laser Doppler vibrometers (LDVs) to detect acoustic echoes from underwater targets[27–30]. However, these optical detection methods lack robustness in uncontrolled

environments and therefore previous demonstrations have been severely limited[27,31]. The presence of ocean surface waves has proven to be a major bottleneck for real-world deployment of such remote underwater imaging systems and is thus a key challenge that needs to be overcome before ubiquitous remote underwater sensing becomes a reality.

To tackle the limitations of existing technologies, we introduce a photoacoustic airborne sonar system (PASS) that leverages the ideal properties of electromagnetic imaging in air and sonar imaging in water[32]. In our previous work, we presented the concept of PASS and demonstrated preliminary two-dimensional (2D) imaging results in hydrostatic conditions[32]. The primary focus of this work is to investigate and overcome the aforementioned fundamental challenge of a hydrodynamic water surface on remote underwater imaging systems such as PASS, thereby opening the door to future deployment in realistic, uncontrolled environments.

As shown conceptually in Fig. 1, PASS generates a remote underwater sound source through the laser-induced photoacoustic effect[33–35]. The laser-generated sound propagates underwater similarly to conventional sonar, reflects from objects in the underwater scene, and in some part propagates back towards the water surface. A small fraction of the sound is able to pass through the water surface into the air where it can be detected by high-sensitivity, air-coupled ultrasound transducers. However, in hydrodynamic conditions, as we will articulate in greater depth in the "Results" section, the non-planar water surface distorts the acoustic echoes as they cross through the air–water interface, thus prohibiting conventional image reconstruction.

In this work, we propose fusion of the PASS imaging modality with three-dimensional (3D) water surface mapping to provide complementary multi-modal inputs to a custom image reconstruction algorithm. Through 3D mapping of the water surface, we obtain a sufficiently accurate model of the acoustic propagation channel such that we can invert the distortion effects caused by a non-planar water surface. By employing this multi-modal sensor fusion framework, we demonstrate experimentally and through simulations that PASS can reconstruct high-fidelity images in hydrodynamic conditions. Lastly, we present in-depth analysis of water surface mapping requirements such that future work can continue to develop PASS into a fully airborne system operating in realistic deployment scenarios.

## Results

**Hydrodynamic conditions**. In hydrodynamic conditions, the air–water interface is non-planar as a result of the water's surface waves; this is in contrast to hydrostatic conditions where the water volume is in a steady state and has a planar surface. An important note for imaging in hydrodynamic conditions is that we can invoke a quasi-static assumption: since the speed-of-sound is significantly greater than the propagation speed of the water's surface waves, the propagation of the surface waves during the data capture can be neglected in most cases (see the "Discussion" section). That being said, the challenge of imaging in hydrodynamic conditions is not the water dynamics but rather the non-planar interface that arises.

In Fig. 2, we contrast acoustic propagation and image reconstruction in hydrostatic and hydrodynamic conditions. In the top row of Fig. 2, the acoustic forward propagation is simulated for a hydrostatic imaging scenario and the image reconstruction accurately recovers the underwater target. In the middle row, a hydrodynamic imaging scenario is simulated which illustrates the distortion (i.e., loss of spatial coherence) that is incurred to the acoustic wavefront as a result of the non-planar

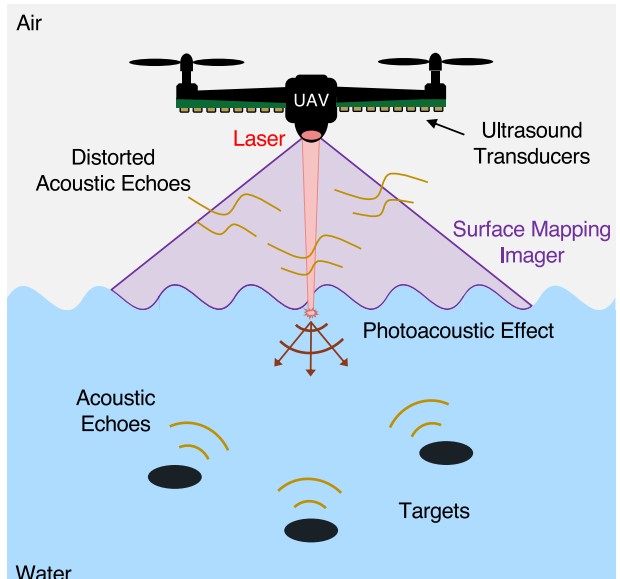

**Fig. 1 Photoacoustic airborne sonar system.** Schematic of proposed system with the laser excitation source, ultrasound detectors, and surface mapping imager all on-board an airborne platform which here is depicted as an Unmanned Aerial Vehicle (UAV).

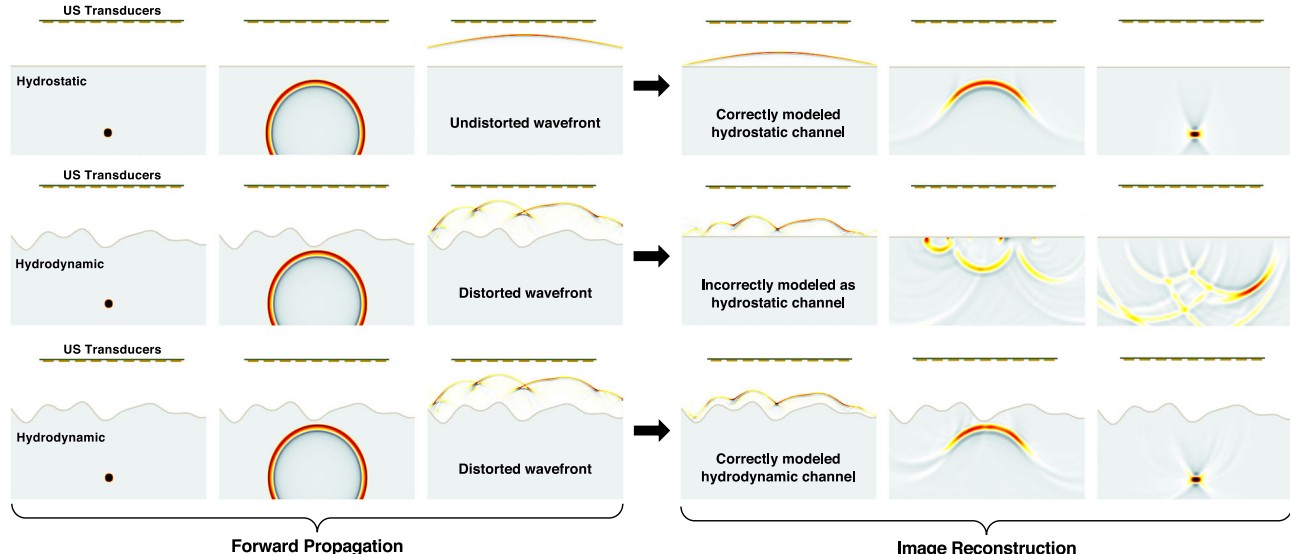

**Fig. 2 Hydrostatic vs. hydrodynamic air–water interface.** Top: Simulated forward propagation and image reconstruction in hydrostatic conditions. Middle: Simulated propagation in hydrodynamic conditions, but assumed hydrostatic in image reconstruction. Bottom: Simulated propagation in hydrodynamic conditions, with correct hydrodynamic channel in image reconstruction.

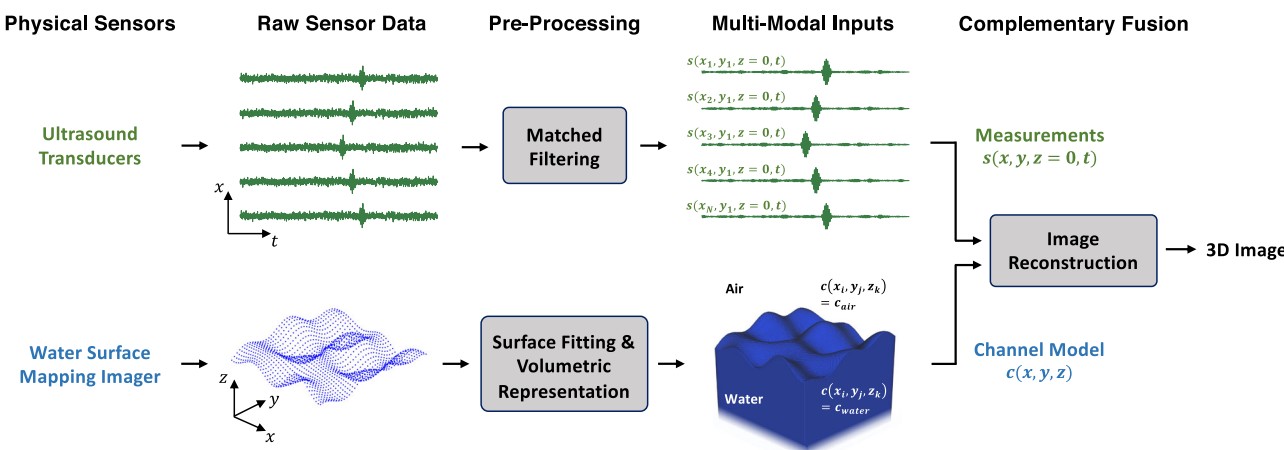

**Fig. 3 Multi-modal sensor fusion.** Ultrasound transducers capture acoustic signals while a 3D imager maps the surface of the water. The raw sensor data are pre-processed to generate the complementary acoustic measurements and channel model which are consumed by an image reconstruction algorithm.

water surface. Consequently, if the image reconstruction incorrectly models a hydrostatic channel, the resulting image will be incoherent. Lastly, in the bottom row, the hydrodynamic imaging scenario is again simulated; however, if the image reconstruction correctly models the hydrodynamic channel, the distortion is compensated and spatial coherence is regained in the water—allowing for successful image reconstruction.

**Multi-modal sensor fusion**. The basis of most coherent image reconstruction algorithms is the idea that the captured signals can be reversed, or migrated, to where they reflected from the scene in order to recover an image. As articulated above in the context of our application, reversing the signals through the water surface, while successfully nullifying the distortion, requires precise knowledge of the water surface profile. In Fig. 3, we propose a multi-modal sensor fusion framework which provides the complementary information required to recover accurately reconstructed images of the underwater scene using PASS.

Here, we discuss the two independent sensor and processing pipelines shown in Fig. 3 before later articulating how these

multi-modal inputs are consumed by the image reconstruction algorithm. In the acoustic pipeline, an ultrasound transducer array and its interfacing electronics convert the airborne sound into electrical signals. The raw sensor data are then matched filtered for optimal noise reduction[36]. As discussed above, the acoustic signals could potentially have been distorted by the existence of a non-planar water surface—necessitating appropriate compensation.

As shown in Fig. 3, we propose that a surface mapping imager which provides 3D spatial information about the water surface profile can be used to capture this complementary input that is required to perform compensation of the distortion. We develop the sensor fusion framework to be general to any imager capable of profiling the water surface; however, in the "Discussion" section we will discuss practical considerations and implementation details for the water surface mapping imager. The raw surface profile, here depicted as a point cloud, is fitted with a continuous surface which could be achieved either via interpolation and filtering or through model-based surface fitting—depending on the density of surface measurements provided by the imager. Next, the surface map is converted into a discretized

3D volumetric representation of the acoustic channel defined over space, $c(x, y, z)$, where voxels above the water surface are assigned the speed-of-sound in air, $c_{air}$, and voxels beneath the water surface are assigned the speed-of-sound in water, $c_{water}$. With this model of the propagation channel, along with the corresponding acoustic measurements, an image reconstruction algorithm can now migrate the signals through the water surface while compensating for the distortions.

**Reconstruction algorithm.** Central to any time-of-flight based imaging system, including acoustic imaging, is the translation of temporal measurements into images by exploiting the propagation speed of the signal through the environment, i.e.:

$$d_{target} = \frac{c_{medium} \cdot t_{TOA}}{2},$$ (1)

where $d_{target}$ is the distance of the target from the imaging system, $c_{medium}$ is the propagation speed of the signal through space, $t_{TOA}$ is the time-of-arrival of the signal, and the factor of one-half comes from two-way propagation that exists in most active imaging systems. This relationship between space and time lends simplicity to image reconstruction in homogeneous media where a global constant value of $c_{medium}$ can be assumed everywhere in space.

On the other hand, imaging in heterogeneous media, for example across the air–water boundary, requires an accurate understanding of the speed-of-sound as a function of space, i.e., $c(x, y, z)$. Above, we referred to $c(x, y, z)$ as the channel model, as it fully encapsulates the required information to understand the relationship between the temporal acoustic measurements captured by the ultrasonic transducers and the unknown target that we desire to reconstruct.

In our previous work, which demonstrated image reconstruction in hydrostatic conditions, we adapted the piece-wise SAR (PW-SAR) algorithm[37] which permits reconstruction in layered media (i.e., heterogeneous, but with planar interfaces) through a piece-wise homogeneous approach. Here, we generalize the piece-wise SAR (GPW-SAR) algorithm such that it can exploit the heterogeneous channel model, $c(x, y, z)$, defined over the reconstruction grid, to perform image reconstruction in hydrodynamic conditions. Similarly to previously developed algorithms[38,39] for imaging through non-planar interfaces, our GPW-SAR algorithm compensates the acoustic signals as they are migrated through the non-planar surface such that any conventional homogeneous image reconstruction algorithm can then be employed.

It should be noted that the GPW-SAR algorithm described below primarily operates in the spectral-frequency domain rather than the space-time domain for computational efficiency; nevertheless, the relationship in Eq. (1) is still central to the underlying physical tie between space and time, or equivalently in the spectral-frequency domain, wavenumber and frequency:

$$k_{medium} = \frac{2\pi f}{c_{medium}},$$ (2)

where $f$ is the acoustic frequency and $k_{medium}$ is the corresponding wavenumber in the propagation medium.

To explain the GPW-SAR algorithm, we will refer to Fig. 4 where the equation numbers refer to those described in the text.

*Step 1:* The acoustic measurements captured by the airborne transducers, $s(x, y, z = 0, t)$, and the spatial distribution of the speed-of-sound, $c(x, y, z)$, are input to the reconstruction algorithm.

*Step 2:* The measurements are transformed from the space-time domain into the spectral-frequency domain through a 3D Fast Fourier transform (FFT) over the $x, y$ spatial dimensions and the $t$

time dimension:

$$S(k_x, k_y, z = 0, f) = \mathcal{F}_{x,y,t}\{s(x, y, z = 0, t)\}.$$ (3)

This decomposition of the spherical wavefronts received in the space-time domain to plane waves in the spectral-frequency domain is known as the Weyl expansion[40].

*Step 3:* The plane waves are migrated to above the water surface ($z = z_1$) through a spectral propagator (phase shift) that follows the proper dispersion relation:

$$(k^i)^2 = (k_x^i)^2 + (k_y^i)^2 + (k_z^i)^2,$$ (4)

where $k^i$ is the acoustic wavenumber in medium $i$ and $k_x^i$, $k_y^i$, and $k_z^i$ are its spatial components. In Eq. (4), $i = a$ refers to the air medium and $i = w$ refers to the water medium. For the spectral propagator in Step 3, the dispersion relation for air is used:

$$S(k_x, k_y, z_1, f) = S(k_x, k_y, z = 0, f) \cdot e^{jk_z^a z_1}.$$ (5)

*Step 4:* The reconstruction grid is discretized along the $z$-axis with voxel size $\Delta z$. The plane waves are migrated one voxel along the $z$-axis using both the dispersion relation for air and for water and then transformed back to the space domain—creating $s_a(x, y, z', f)$ and $s_w(x, y, z', f)$, respectively:

$$s_a(x, y, z', f) = \mathcal{F}_{x,y}^{-1}\left\{S(k_x, k_y, z_1, f) \cdot e^{jk_z^a \Delta z}\right\},$$ (6)

$$s_w(x, y, z', f) = \mathcal{F}_{x,y}^{-1}\left\{S(k_x, k_y, z_1, f) \cdot e^{jk_z^w \Delta z}\right\}.$$ (7)

*Step 5:* The wavefronts are recombined in the space domain while keeping $s_a(x, y, z', f)$ for air voxels and keeping $s_w(x, y, z', f)$ for water voxels in the modeled acoustic channel:

$$s(x, y, z', f) = \gamma_a s_a(x, y, z', f) + \gamma_w s_w(x, y, z', f),$$ (8)

where $\gamma_a(x, y, z') = 1$ where $c(x, y, z') = c_a$ and where $\gamma_w(x, y, z') = 1$ where $c(x, y, z') = c_w$. The recombined wavefront is transformed back to the spectral domain before repeating Step 4 and Step 5 for all discretized depths between $z_1$ and $z_2$.

*Step 6:* The remainder of the algorithm is simply a homogeneous image reconstruction problem. The plane waves are propagated to each depth $z$ and transformed back to the spatial domain.

*Step 7:* The complex reconstructed image $\Gamma(x, y, z)$ is formed by summing over all frequencies:

$$\Gamma(x, y, z) = \sum_f \mathcal{F}_{x,y}^{-1}\left\{S(k_x, k_y, z_2, f) \cdot e^{jk_z^w(z-z_2)}\right\} \cdot e^{jk_w r},$$ (9)

where the final phase term in Eq. (9) compensates for the phase that was accumulated from the location of the acoustic source, $(x_s, y_s, z_s)$, to the scene where:

$$r = \sqrt{(x - x_s)^2 + (y - y_s)^2 + (z - z_s)^2}.$$ (10)

Lastly, a final image can be displayed by taking the magnitude of the complex image: $|\Gamma(x, y, z)|$.

An interesting note is that it is not required to explicitly account for refraction as we migrate the signals through the air–water interface in Steps 4–5. This is another advantage of using a spectral propagator in the spectral-frequency domain rather than a spatial propagator in the space-time domain as refraction is inherently handled by the transition of dispersion relation (i.e., $k_z^a$ vs. $k_z^w$) as we cross the air–water interface.

Finally, it should be noted that the presented GPW-SAR algorithm is equivalent to the PW-SAR algorithm when $z_1 = z_2$, i.e., when the water surface is planar. It is clear from the proposed algorithm that without the complementary sensor inputs, the acoustic migration through the non-planar water surface would not be possible.

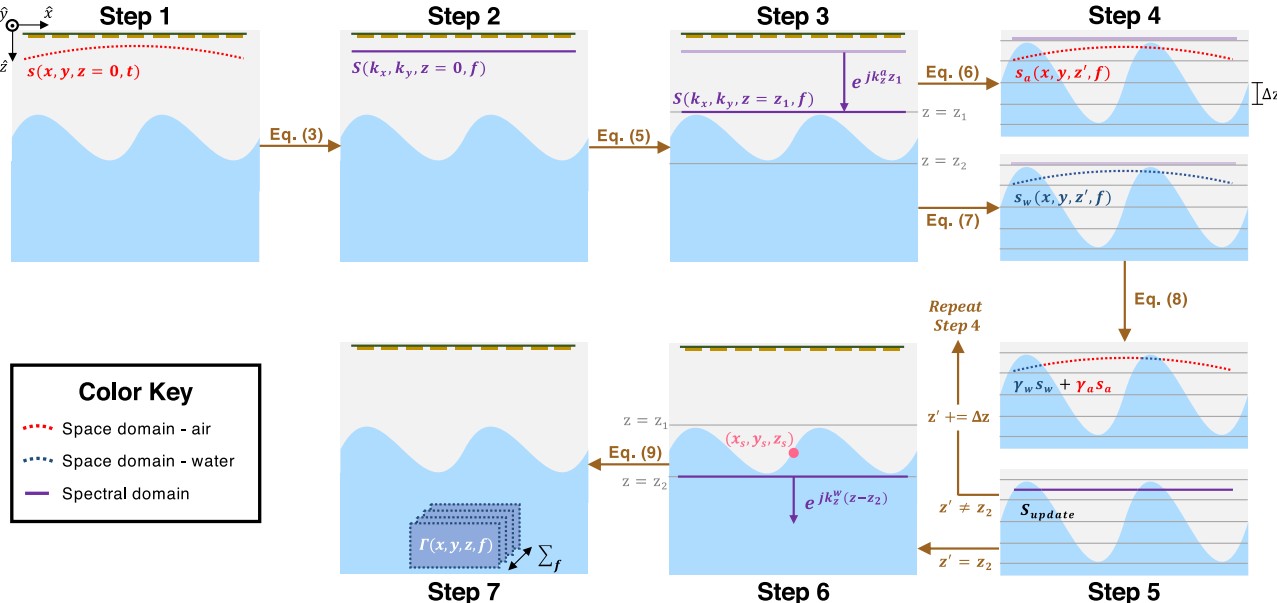

**Fig. 4 Image reconstruction algorithm.** Algorithm steps and equation numbers align with the text. A color key is provided to differentiate between the space domain and the spectral domain in each medium.

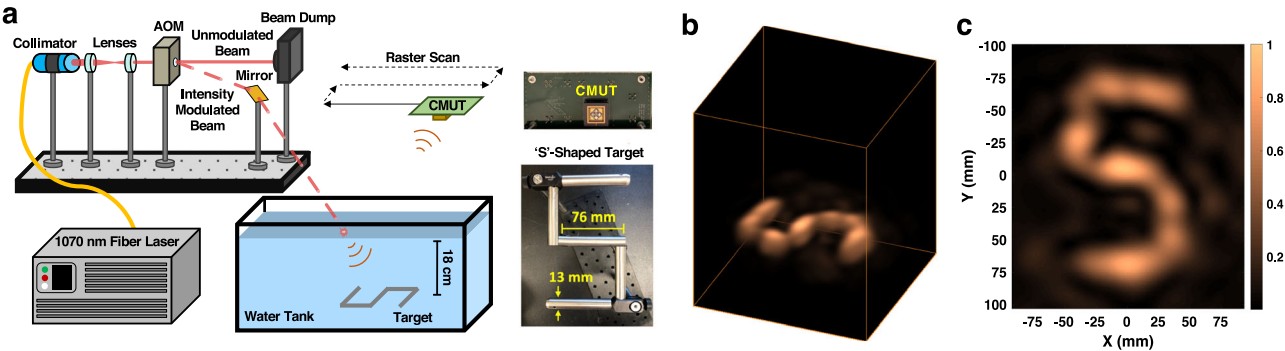

**Fig. 5 Imaging in hydrostatic conditions. a** Schematic of the experimental setup for a fully airborne implementation of the photoacoustic airborne sonar system (PASS) where the laser is intensity modulated using an acousto-optic modulator (AOM), the acoustic echoes are detected by a capacitive micromachined ultrasound transducer (CMUT), and the embedded target is 'S'-shaped. **b** 3D reconstructed image. **c** Bird's-eye view of the reconstructed image, i.e., a depth slice of the reconstructed volume at the target depth.

**3D imaging in hydrostatic conditions.** First, we expand on the results of our previous work by experimentally demonstrating 3D imaging results using a fully airborne (i.e., end-to-end) proof-of-concept implementation of PASS in hydrostatic conditions. A schematic depiction of the lab-based setup is shown in Fig. 5a. A burst of infrared light is fired from a quasi-continuous-wave laser. The free-space laser beam is coupled through an acousto-optic modulator (AOM) which modulates the laser burst at the desired acoustic frequency. The modulated laser beam strikes a mirror which reflects the beam towards the water surface, where it is absorbed. The laser-generated underwater acoustic signals are then incident on the depicted 'S'-shaped target and are reflected back toward the water surface. In this hydrostatic experiment, the acoustic echoes pass through the planar water surface without incurring distortion and are detected by a custom, high-sensitivity capacitive micromachined ultrasound transducer (CMUT).

To achieve high sensitivity and resilience to noise, the airborne CMUT used in our experiments is a resonant device with a 71 kHz resonance frequency and only 3 kHz of bandwidth[41]. The modulation of the laser intensity by the AOM dictates the frequency at which the sound waves are generated by the photoacoustic effect; therefore, we strategically modulate the laser

intensity to maximize the acoustic energy at the CMUT's resonance frequency to ensure efficient detection[42].

After the sound is detected, the interfacing electronics amplify, filter, and digitize it into a signal that is passed to a digital signal processing pipeline. In order to convert detected signals into a reconstructed image, spatial information must be obtained by capturing the airborne sound over an aperture—either with a physical array of transducers or, as implemented here, raster scanning a single transducer to form a synthetic aperture.

In previous work, we presented 2D imaging results obtained by scanning the CMUT over a one-dimensional synthetic aperture[32]. In this work, we demonstrate for the first time 3D imaging of an underwater scene which requires scanning the CMUT over a 2D aperture while repeating the acoustic data capture at each location; raster scanning a single transducer with coherent detection effectively mimics simultaneous detection with an array of transducers.

By employing the PW-SAR algorithm, we are able to compute the 3D image displayed in Fig. 5b. A 2D depth slice, or cross-section, of the bird's-eye view is shown in Fig. 5c for easy comparison of the reconstructed image to the ground-truth target.

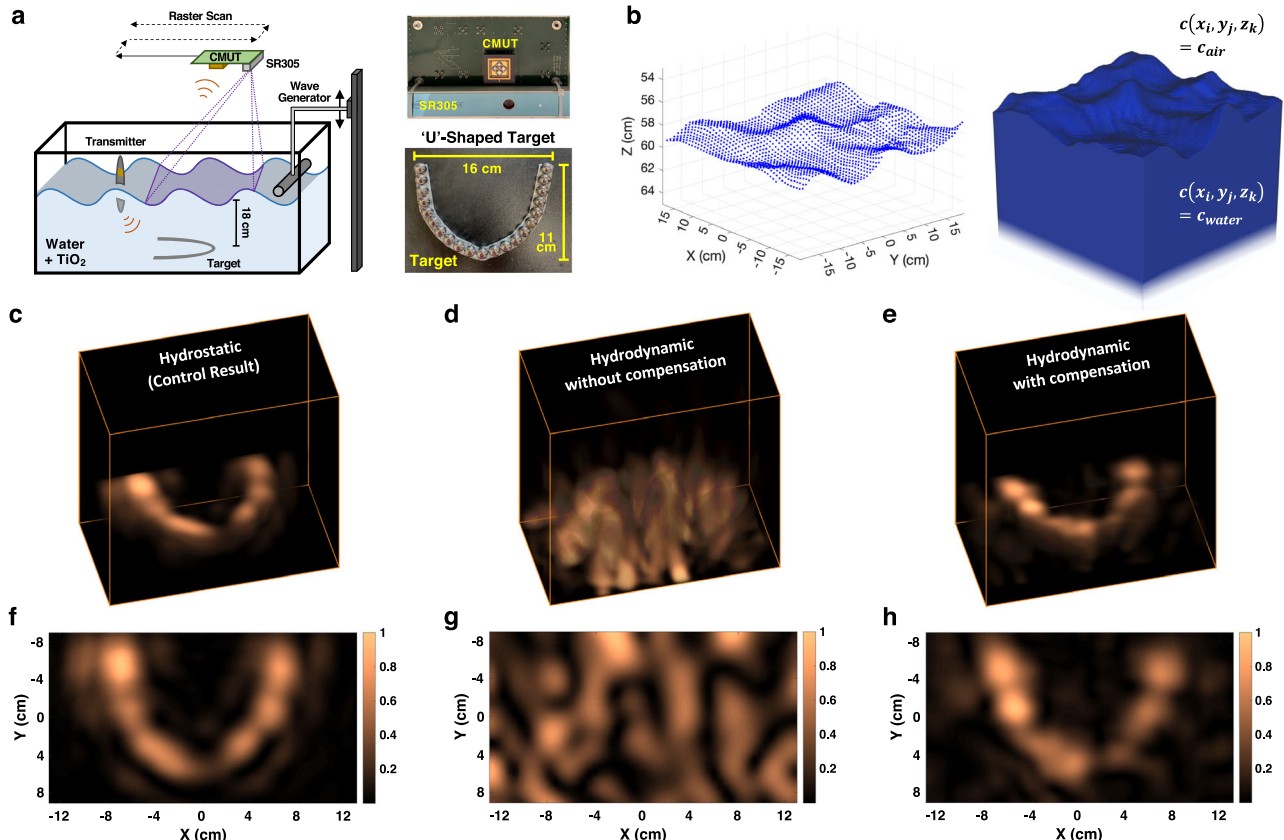

**Fig. 6 Imaging in hydrostatic and hydrodynamic conditions. a** Schematic of the experimental setup where an acoustic transmitter replaces the laser excitation, a capacitive micromachined ultrasound transducer (CMUT) detects the acoustic echoes, a depth sensor (SR305) profiles the water surface, and the embedded target is 'U'-shaped. **b** Example point cloud captured by the SR305 and the corresponding channel model. **c** 3D reconstructed image in hydrostatic conditions. **d** 3D reconstructed image when surface waves are present but are not compensated. **e** 3D reconstructed image when surface waves are present and are properly modeled using the depth sensor. **f**–**h** Bird's-eye view of the reconstructed images in (**c**–**e**).

In this section, we present the first 3D image captured using a fully airborne sonar system by exploiting a laser-generated, remote sound source and air-coupled ultrasonic transducers. The system concept demonstrates high-resolution images and promises scalability to greater depths (see Supplementary Note 1) as well as flexibility for use in various applications[32]. However, before deployment in real-world settings, there are a few remaining challenges to be solved—a major one of which is imaging in hydrodynamic conditions, for which we present promising results in the next section.

**3D imaging in hydrodynamic conditions**. In this section, we experimentally validate the multi-modal sensor fusion framework and GPW-SAR algorithm while exhibiting PASS's imaging capabilities in hydrodynamic conditions. To do so, we must make a few alterations to the fully airborne PASS experimental setup employed in hydrostatic conditions above. A schematic depiction of the experimental setup is shown in Fig. 6a. To enforce a hydrodynamic condition, a wave generator is used to continuously plunge a plastic cylinder in and out of the water.

To map the water surface profile, we use a commercially available coded light depth sensor (Intel RealSense SR305). The coded light depth mapping technology maps the water surface by projecting a series of patterns (coded light) and evaluating the deformation of these patterns caused by the 3D surface[43]. We choose to use a coded light depth sensor for the proof-of-concept implementation due to its superb accuracy (≈1 mm), spatial resolution (≈1 mm), and frame rate (60 Hz) at the expense of robustness in outdoor lighting conditions. Due to poor optical

reflectivity, it is also required to introduce an additive in the water to increase the reflectivity of the infrared light patterns off the normally transparent water surface. For this preliminary demonstration, we use titanium dioxide ($TiO_2$), which when mixed with the water remains suspended and effectively dyes the water white[44]. See the Supplementary Movie for an experimentally captured time-varying surface wave using the SR305 coded light depth sensor. As discussed in detail in the "Discussion" section, future work will focus on developing a surface mapping imager that does not require an additive in the water.

To mitigate the hazard of spurious optical reflections from the dynamic, and highly reflective, $TiO_2$-dyed water surface in a laboratory setting, an acoustic transmitter is placed at the surface of the water to act as a proxy for the laser-generated sound source. The transmitted signal is designed with characteristics that mimic the laser-excited source of the previous hydrostatic experiments. In addition, prior work[31,45] and simulations (see "Discussion" section) further validate that this substitution of the acoustic in-water transmitter in place of laser excitation in hydrodynamic conditions is consistent. The focus of our experiment in hydrodynamic conditions is to, thus, de-risk the fundamental challenge of the airborne acoustic detection pipeline that receives distorted echoes from underwater targets.

The underwater target for this experiment is a metallic 'U'-shaped object. Similarly to the hydrostatic experiment above, the transmitted acoustic signal reflects from the underwater target, propagates through the water surface and is detected by the airborne CMUT. Simultaneously, the coded light depth sensor acquires a map of the water surface. An example raw point cloud

obtained by the depth sensor is shown in Fig. 6b along with the processed channel model that is passed to the image reconstruction algorithm.

First, we image the target in a hydrostatic condition using the modified experimental setup which serves as the control result for further experiments. The target is reconstructed using the PW-SAR algorithm and a high-fidelity image is obtained as shown in Fig. 6c.

Next, we introduce the hydrodynamic conditions and repeat the measurement capture. Employing the PW-SAR algorithm, i.e., not compensating for the non-planar surface, we reconstruct the garbled image shown in Fig. 6d. It is evident that without proper modeling of the acoustic channel, the target reconstruction no longer resembles the control result, as was also observed in Fig. 2.

Finally, if we employ the proposed multi-modal sensor fusion framework and custom GPW-SAR algorithm, we reconstruct the image in Fig. 6e. In addition to the 3D reconstructions shown in Fig. 6c–e, a depth slice at the target's depth is shown in Fig. 6f–h. The high similarity of the 3D image reconstructed in hydrodynamic conditions with the control result captured in hydrostatic conditions demonstrates the efficacy of the proposed solution to compensate the acoustic distortions and maintain the ability to acquire high-fidelity images even in the presence of water surface waves.

The evident robustness in these lab-based proof-of-concept experiments in hydrodynamic conditions marks a major milestone and builds confidence that this framework could be applied to a fully airborne implementation of a photoacoustic airborne sonar system such that it could successfully operate in open, uncontrolled ocean waters.

## Discussion
The proposed PASS imaging modality leverages the photoacoustic effect to remotely generate an underwater sound source and high-sensitivity CMUTs to detect the acoustic echoes in air. We validate the 3D imaging capabilities of the system in controlled, hydrostatic scenarios and—through a modified experimental setup—demonstrate 3D imaging in hydrodynamic conditions. In hydrodynamic conditions, we overcome the distortion of the acoustic signals caused by the non-planar air–water interface through a multi-modal sensor fusion framework. We propose that by mapping the water surface, we can create a model of the acoustic propagation channel such that we can invert the distortion caused by the non-planar water surface through a custom GPW-SAR image reconstruction algorithm.

In the remainder of this section, we (1) revisit the substitution of the underwater acoustic transmitter for the laser-generated source, (2) provide in-depth analysis of the specifications for the next-generation surface mapping solution, and (3) summarize the future work that must be completed so that PASS can employ the presented sensor fusion framework in real-world conditions.

**Transmitter vs. laser-generated source**. As discussed above, due to the safety concerns of spurious optical reflections, particularly when $TiO_2$ is added to the water, we replaced the laser-generated acoustic source with an in-water acoustic transmitter for the hydrodynamic experiments; without addition of $TiO_2$, reflections would not be a concern. Previous works have analytically solved for and experimentally verified the generated underwater photoacoustic signal as a function of several parameters—including the laser intensity modulation function and incidence angle on the water surface[32,45]. Using this analytical solution, we designed the transmitted acoustic signal with modulation and pressure

level that closely matches the laser-generated photoacoustic signal, more details of which are provided in "Methods" section.

Since the underwater acoustic signal is effectively the same whether it is laser-generated or generated by an acoustic transmitter, the validity of this substitution in our experiments is dependent solely on the impact of the laser having oblique incidence on a non-planar water surface. Using the analytical solution to the photoacoustic effect, we have performed verifying simulations. Fig. 7b, c compare the directivity of the insonification of a laser-generated acoustic source for normal versus oblique incidence. With sufficiently small laser beam radius (relative to the acoustic wavelength in water), the directivity is nearly hemispherically isotropic[32]. As depicted in the figure, with oblique incidence, the field-of-view of the source shifts with the angle of incidence, though because it is nearly hemispherically isotropic, there is little impact on the underwater insonification—especially at greater depths.

Several researchers have studied the statistics of water waves and have found that the local slopes of the waves follow a Gaussian distribution where 95% ($2\sigma$) are less than 20° in reasonable wind conditions[46,47]. In Fig. 7a, we plot the normalized pressure at $\psi = 0°$, or along the depth-axis beneath the point of laser absorption, as a function of the laser incidence angle. This plot illustrates that there is insignificant change in the amplitude of the sound source that propagates toward the depths of the water, even at incident angles as large as 40°, the maximum statistically significant wave slope[47]. All considered, the use of the in-water isotropic acoustic transmitter serves as a valid substitute for proof-of-concept experiments.

**Surface mapping alternatives**. In the hydrodynamic experiments presented herein, we used a commercially available coded light depth sensor for 3D water surface mapping. This required an additive to the water to increase the reflectivity of the coded light patterns from the water surface. In practice, it is not feasible to introduce additives to the water, so it will be critical to develop a surface mapping solution that has the accuracy, spatial resolution, and frame rate that will enable the PASS imaging modality to exploit the presented multi-modal sensor fusion framework in real-world deployment.

In addition to coded light[48], other depth sensing technologies have been used to map the water surface including scanning lidar[49], stereo imaging[50], polarimetric imaging[51], and ultrasonic sensing[52]. A future implementation of our system toward deployment in open waters may utilize one, or a combination of, these techniques to map the surface of water. To understand the specifications and the trade-off space for the design of the next-generation surface mapping solution, we analyze the requirements demanded by PASS below.

**Surface mapping accuracy**. In order to establish the surface mapping accuracy requirement, we developed a custom acoustic forward simulator for heterogeneous media that exploits a technique known as the Hybrid Angular Spectrum Method[53]. The simulator models PASS by encapsulating the 3D acoustic propagation from the laser-generated sound source, reflection from a point target at a prescribed underwater depth, transmission across a non-planar air–water interface, and finally to airborne detection with transducers at a prescribed height.

An example simulation setup and hydrodynamic surface profile are shown in Fig. 8a, b. To evaluate the impact of surface mapping errors (i.e., inaccuracy), we first simulate the forward propagation using the ground-truth surface map and then add spatially filtered, normally distributed errors to the ground-truth surface map prior to employing the GPW-SAR algorithm for

reconstruction. The spatial filtering ensures that the errors have low spatial frequencies that are expected in water surface waves.

To quantify the impact of surface map errors, we compute the normalized cross-correlation ($NCC$) of the reconstructed image relative to the reconstructed image computed using the ground-truth surface map. The $NCC$ characterizes the degradation in image quality as a function of surface map errors, with an $NCC$ close to one corresponding to minimal degradation. We simulate the effect of surface map errors as a function of several different parameters: (1) the acoustic frequency ($f$) of the sound source, (2) the height ($H$) of the receivers above the mean water surface, (3) the depth ($D$) of the target in the water, and (4) the peak-to-peak amplitude of the water surface waves.

The results of the simulations are shown in Fig. 8c–f. For each of the parameters of interest, we perform the reconstruction with increasing level of surface map errors, such that the root mean square error (RMSE) with the ground truth varies from 0 mm to 5 mm. For each of the plots, five simulations with the same parameters, but with different surface profiles, were conducted

and then averaged; the same five surface profiles were used across all plots. This was to ensure the trends were consistent across several different surface profiles.

Figure 8c shows a high-correlation between acoustic frequency and degradation in image quality with increased RMSE; as the frequency increases, the errors become a larger fraction of the acoustic wavelength and thus have a greater impact. In Fig. 8d, there exists correlation between height of the receivers and image quality with higher heights showing lesser image degradation for the same level of error. In Fig. 8e, there is little-to-no correlation between the depth of the target and image quality degradation; out of the five surface profiles simulated, there is no consistent trend. Lastly, in Fig. 8f, larger amplitude surface waves demonstrate lesser image degradation with increased RMSE likely due to the fact that the errors are a smaller fraction of the overall wave height. It should be noted that computational constraints limited the maximum receiver height and target depth that could be simulated to 3 m; however, it is expected that these trends hold for larger distances.

In conclusion, the simulations informed that PASS requires millimeter-scale surface mapping accuracy, with this requirement being more stringent for high acoustic frequencies and small wave heights.

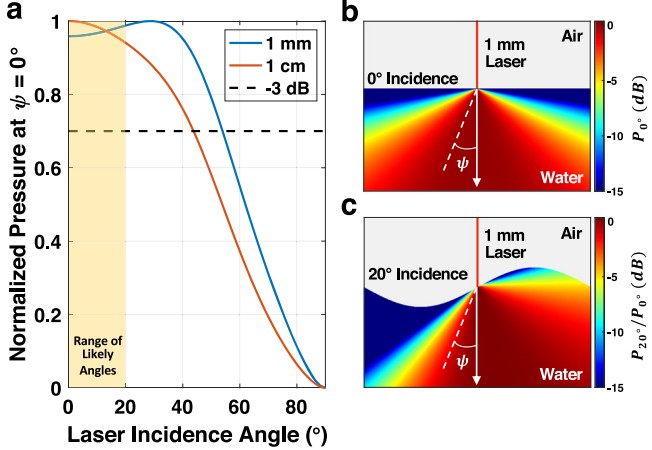

**Fig. 7 Oblique laser incidence. a** Normalized pressure along the depth-axis ($\psi = 0°$) as a function of the laser incidence angle for laser beam radii of 1mm and 1 cm. **b** Underwater acoustic directivity for normal incidence. **c** Underwater acoustic directivity for oblique incidence.

**Surface mapping spatial resolution.** Due to the spatial periodicity of water waves, a surface map at a desired resolution can be restored if we spatially Nyquist sample the water surface. To further articulate, open water waves, driven mostly by wind and gravity, can be decomposed into a superposition of waves with different wavelengths or spatial frequencies. In an attempt to model the spectrum of ocean waves, several researchers have characterized the statistics of wave energy as a function of wavelength[54]. Capillary waves, waves with wavelengths smaller than approximately 2 cm, have amplitudes that are often insignificant under reasonable wind conditions[55].

Therefore, if we make the assumption that wavelengths less than 2 cm have negligible amplitude, we can effectively Nyquist sample with a surface mapping technology that has 1 cm spatial resolution or better. That said, it is possible that the spatial periodicity of water waves could be leveraged to enable other non-uniform sampling schemes. With spatial Nyquist sampling, a surface profile with arbitrarily high resolution can be restored

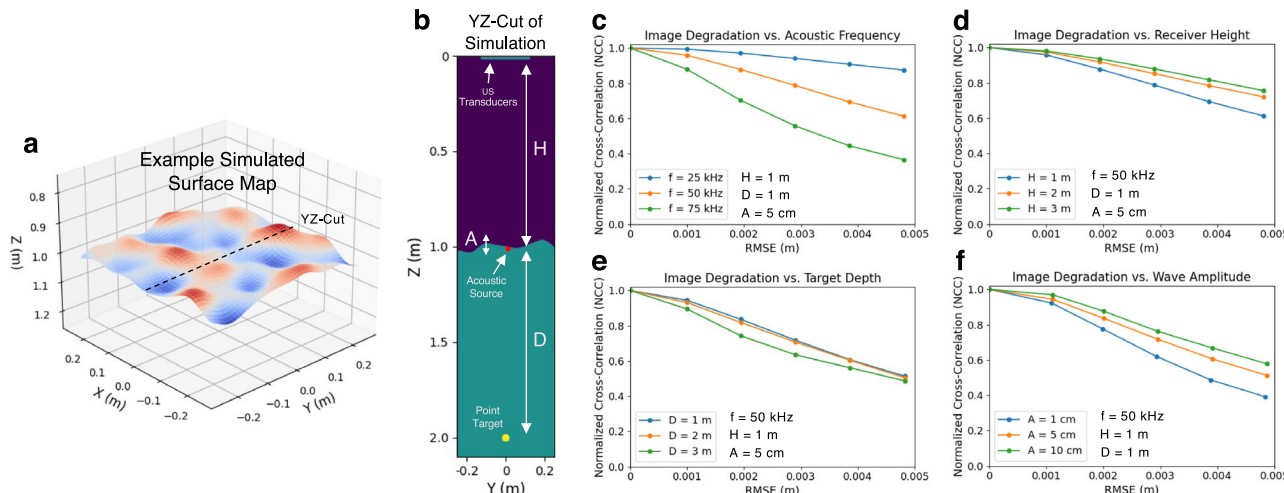

**Fig. 8 Surface mapping accuracy simulations. a** Example surface map used in the simulations. **b** YZ-Cut of the example simulation setup showing the acoustic source at the water surface with waves that have peak-to-peak amplitude A, underwater target at depth D, and ultrasound (US) transducers at height H. Simulated image degradation with surface map errors as a function of: **c** acoustic frequency, **d** receiver height, **e** target depth, and **f** wave amplitude.

through appropriate interpolation in order to match the desired resolution of the computational grid used in the image reconstruction algorithm. For the GPW-SAR algorithm presented in this work, we have found that the spatial resolution $\Delta x = \Delta y = \lambda_{air}/4$ provides a good balance between computational complexity and accuracy.

**Surface mapping frame rate.** Lastly, to identify the necessary frame rate, we analyze the propagation of the water's surface waves. Previously, we mentioned that we can neglect the propagation of the surface waves during an acoustic measurement due to the relatively high speed of the acoustic signals in comparison to the surface waves. This concept is similar to channel coherence time in communications systems[56]. If the acoustic signals of interest are within a single coherence interval, i.e., they span short distances (as they did in our experiments), the quasi-static assumption holds. On the other hand, if the acoustic signals span long distances, for example if there is a shallow target and a deep target, the packet of acoustic echoes from the deep target may encounter surface waves that have since propagated from when the acoustic echoes from the shallow target crossed into air. In this case, a single surface map may not suffice and successive surface maps would need to be captured and assigned to subdivided coherence intervals of the received signals.

Fortunately, the temporal periodicity of water waves can be exploited to achieve arbitrarily high effective frame rates through systematic interpolation of lower frame rate surface mapping. In the following analysis, we calculate the minimum possible frame rate that permits unambiguous interpolation of the surface mapping frames. As mentioned above, open water waves can be decomposed into a superposition of waves with different wavelengths or spatial frequencies. In Fig. 9, we illustrate a spectral decomposition of a simplified 2D surface wave—though the same process can be translated to 3D surface profiles comprised of more wavelengths. The wave height at time $t$ and position $x$, denoted $h(x,t)$, can be written as the superposition of $N$ monochromatic waves:

$$h(x,t) = \sum_{i=1}^{N} a_i \sin(k_i x - \omega_i t + \psi_i), \qquad (11)$$

where $k_i$ is the wavenumber $2\pi/L_i$ of the $i$-th wave component with wavelength $L_i$, and where $\omega_i$ is the angular frequency, $\psi_i$ is the initial phase, and $a_i$ is the amplitude of the $i$-th wave component. The phase change as a function of time is therefore:

$$\Delta \phi_i = \omega_i \Delta t, \qquad (12)$$

where the angular frequency in open water waves is a function of the wavenumber and the gravitational acceleration constant $g = 9.81$ m/s[57]:

$$\omega_i = \sqrt{k_i g}. \qquad (13)$$

As shown in Eq. (13) and Fig. 9, smaller wavelengths have higher angular frequencies and equivalently have faster phase accumulation. Consequently, to avoid phase ambiguity, the surface mapping frame rate must be high enough such that $\Delta \phi < \pi$ for the smallest wavelength of interest, or $L = 2$ cm. Therefore, from Eq. (12) and Eq. (13), the minimum frames per second (FPS) is:

$$FPS_{min} = \frac{1}{\Delta t_{max}} = \frac{\sqrt{9.81 \cdot 2\pi/0.02}}{\pi} \approx 18 \text{ FPS}. \qquad (14)$$

If this minimum frame rate is exceeded by the surface mapping imager, unambiguous interpolation could be employed to achieve a higher effective frame rate.

**Future work.** Now that we have established specifications for a surface mapping solution, future work will involve developing a surface imager that is robust in outdoor lighting conditions and is capable of millimeter-scale accuracy, spatial resolution of ≤1 cm, and frame rate of ≥18 FPS—without introducing an additive to the water. In addition, future work could involve exploring computational approaches that reduce the demands of the surface mapping imager. Future work will also analyze second-order wave effects, such as sea spray, whitecaps, large swells, etc. that may require additional mitigation strategies. Finally, we will take the next steps of demonstrating this proof-of-concept photo-acoustic airborne sonar system in real-world conditions by employing the developed multi-modal sensor fusion framework presented herein.

## Methods

**Hydrostatic experimental setup.** A 100 μs burst with 2.7 kW peak power (<10 W average power) is output from a fiber laser operating with a 1070-nm wavelength (IPG Photonics YLR-450/4500-QCW-AC). The laser wavelength was chosen using the analysis outlined in our previous work[32]. The burst is coupled into an AOM (Gooch & Housego AOMO 3095-199) which has approximately a 25% modulation efficiency (diffraction efficiency and insertion loss); this efficiency is low due to the incoherence of the available laser. The applied intensity modulation function employs a previously published coded pulse encoding technique[42,58]—here we use 3 excitation pulses and 2 suppression pulses with a 71 kHz modulation frequency. The diffracted output of the AOM is therefore intensity modulated with approximately 675 W peak power (<2 W average power). The diffracted beam deflects from a mirror toward the water surface, where it is absorbed. The diameter of the laser beam at the point of absorption is less than 1 mm, which we have shown generates a nearly hemispherically isotropic sound source in the water[32]. The estimated source pressure level for the laser-generated source is ≈1 Pa or 120 dB re. 1 μPa at 1 m distance from the source.

The underwater target is propped up from beneath such that it sits at an 18 cm depth. The target is constructed from metal rods and is in the shape of an 'S'; the rods are 76 mm in length and 13 mm in diameter. The dimensions of the water tank are 60 cm × 50 cm × 30 cm ($L \times W \times D$). The water in the tank is not disturbed by external forces and therefore is in a hydrostatic state.

The airborne CMUT operates at a resonance frequency of 71 kHz and is interfaced with in-house analog front-end electronics consisting of a low-noise transimpedance amplifier and additional voltage amplification and filtering stages. The signal is then digitized by an oscilloscope and is read into a computer. The CMUT, which is at a standoff of approximately 20 cm from the water, is scanned using linear translation stages in increments of $\lambda$ (4.8 mm) over a 24 cm by 20 cm aperture. At each location, the measurement is repeated. Temporal synchronization of measurements is performed by starting each acoustic measurement at the time of the laser burst; this ensures coherent detection across the scanned aperture. An image of the 'S'-shaped target is reconstructed using the PW-SAR algorithm, which assumes that the speed-of-sound in air is 340 m/s and the speed-of-sound in water is 1500 m/s.

**Hydrodynamic experimental setup.** To closely mimic the hydrostatic experiment, the acoustic transmitter (RESON TC4034) is programmed to transmit an acoustic signal that matches the expected acoustic source generated by the coded pulse modulated laser. The estimated source pressure level of the transmitter is ≈3 Pa or 130 dB re. 1 μPa at 1 m distance from the source. The transmitter, placed just beneath the surface, is used as a proxy for the laser excitation to eliminate the hazards of uncontrolled optical reflections from the dynamic water surface in a lab environment—particularly when the TiO₂ is added. The transmitter is placed towards the edge of the water tank so as to not obstruct the acoustic echoes from propagating into the air.

The underwater target again sits at an 18 cm depth, although this time in the shape of a 'U' to differentiate the two experiments. The target is 16 cm × 11 cm and is constructed from metal spheres each with a 13 mm diameter. For the hydrodynamic experiments, the water is consistently disturbed during the measurement capture by a plunging plastic cylinder. The peak-to-peak amplitude of the waves in the experiments is on the order of 3–5 cm.

Similarly to the hydrostatic experiments, the CMUT is scanned over a 2D aperture while capturing a measurement at each location. For this experiment, the CMUT is 60 cm above the mean water surface and the scanned aperture is 26 cm × 24 cm. The measurements are temporally synchronized with the signal transmission, as before, although here coherence is not maintained across the scanned aperture due to the hydrodynamic channel. Consequently, we use a coded light depth sensor (Intel Realsense SR305) to capture a map of the water surface at each measurement location. The depth sensor is aligned adjacent to the CMUT with a known fixed offset. For each location, the surface map over a 26 cm × 24 cm region-of-interest (ROI) in the water tank is extracted, ensuring that the ROI is consistent across every measurement. The processing of raw surface maps acquired

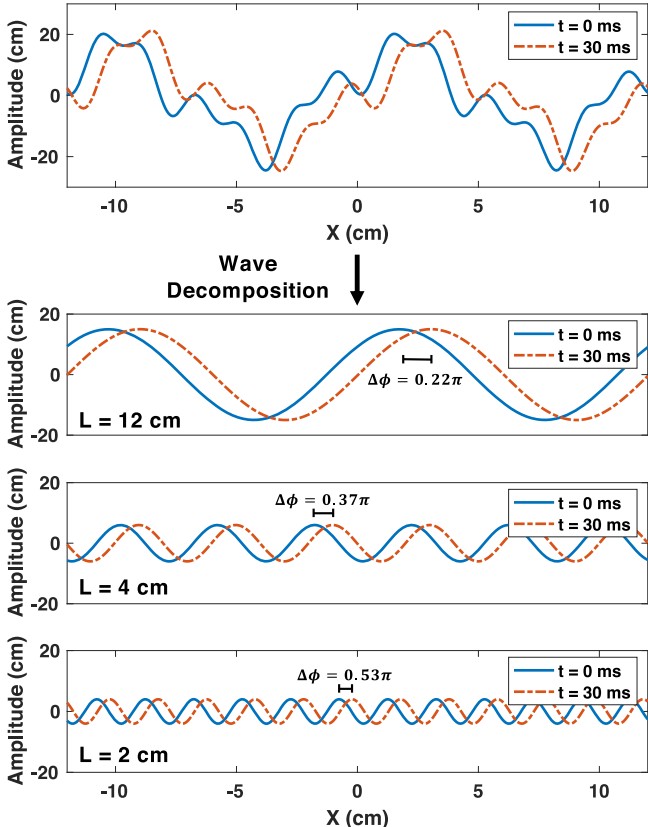

**Fig. 9 Surface wave decomposition.** Decomposition of waves to analyze the phase accumulation ($\Delta\phi$) of each wavelength ($L$) at a given time ($t$). Here, we illustrate the propagation of wave components with a 12 cm, 4 cm, and 2 cm wavelength over a period of 30 ms.

by the depth sensor into channel models consumed by the reconstruction algorithm is outlined in more detail in Supplementary Note 2.

Unlike for the hydrostatic imaging experiments, the image reconstruction cannot be performed over all measurements simultaneously due to the lack of temporal coherence over the synthetic aperture. Instead, each measurement must be individually migrated to the underwater scene with a corresponding channel model for each acoustic measurement. Therefore, by performing the reconstruction procedure on individual measurements and coherently adding the resulting images, the final reconstructed image successfully depicts the scene. The reconstruction procedure for the hydrodynamic experiment is therefore summarized by:

$$I(x,y,z) = \sum_{i=1}^{N} \text{GPW} - \text{SAR}(s(x_i, y_i, z=0, t), c_i(x,y,z)), \quad (15)$$

where $N$ is the number of scan locations in the synthetic aperture and where $s(x_i, y_i, z=0, t)$ and $c_i(x,y,z)$ are the acoustic measurement and the speed-of-sound channel model at each scan location. In the channel model, the speed-of-sound in air is assumed to be 340 m/s and the speed-of-sound in water is assumed to be 1500 m/s. Supplementary Note 3 discusses the impact of improperly assumed speed-of-sound on the reconstructed image.

It should be noted that in practical deployment, an ultrasound transducer array would be utilized such that only a single surface map would need to be captured for the array of ultrasonic measurements. In this case, the full field-of-view of the surface mapping imager (rather than a smaller ROI) could be utilized and the GPW-SAR algorithm could be applied directly.

## Data availability

The data that support the findings of this study are available from the corresponding author upon reasonable request.

## Code availability

The code that supports the findings of this study are available from the corresponding author upon reasonable request.

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

## Acknowledgements

The authors would like to thank Prof. B. T. Khuri-Yakub and his research group for the design and fabrication of the utilized CMUTs.

## Author contributions

A.F., A.S., and A.A. conceived the system and methods. A.F. and R.P.M. performed analysis and simulations to determine implementation details. A.F. and R.P.M. developed the processing pipeline for surface mapping. A.F. developed the image reconstruction algorithm and performed the experiments. A.S. designed the transducer's interfacing electronics. All authors contributed to the manuscript.

## Competing interests

The authors declare no competing interests.
