## [Peer Review File · Communications Engineering]

Reviewers' comments:

Reviewer #1 (Remarks to the Author):

This manuscript proposes to demonstrate 3-D sonar imaging of underwater targets using an airborne system. After a rather long introduction to show that underwater sonars are not up to the task, there is a short description of the current state of photoacoustics. This is followed with a thorough description of the results and methodology, with the presentation of image reconstruction from tank experiments over an area of 12 by 16 cm (Figure 5), imaged from 60 cm above the water. These experiments are done with the addition of titanium dioxide in the water, and an embedded target constructed from metallic spheres, emplaced 18 cm deep. The addition of centimeter-high waves makes the task more complex, and the article nicely builds up on 2-D hydrostatic tests by the authors (reference 17).

The technical achievements are promising but I do not think this article is at the level expected of Nature Communications Engineering. I am therefore reluctantly arguing for major revisions, not because of the technical achievements but because of the way they are presented.

This includes the absence of explanations of how any innovation would stand up in the application of sonar imaging. Sonar imaging typically covers areas several kilometres square, depths ranging from centimetres to kilometres, and objects that are much more challenging to detect than metallic spheres (less acoustic differences with their backgrounds). Water bodies do not have the benefit of added titanium oxide either. So, at the very least, I would have expected an explanation of how these lab experiments in a very constrained setting would apply to real-world applications.

The authors are doing very good experimental work, and I remember enjoying their 2020 IEEE article. It nicely presented the use of capacitive micromachined ultrasonic transducers (CMUTs), it was backed up by a stronger and more relevant bibliography, and the general concept was better presented, e.g. in its Figure 1. Measurements of acoustic attenuation (e.g. Figure 7) and other constraints of airborne acquisition on acoustic imaging made these results very useful. The extension to 3-D and hydrodynamic conditions needs to do the same. The Supplementary Material provides relatively similar information, for example that some targets could be detected down to 100 m deep, when imaging at 50 kHz, with close to 18 frames per second, but this is not directly presented and not fully discussed.

These results could also have advantageously been presented in the Abstract, currently devoid of any numerical information.

The current bibliography presents some interesting articles but does not achieve the breadth of the references in the IEEE 2020 article, and it was interesting to see no reference to some of the pioneering photoacoustic work of V. Humphrey et al. (from the 1980s onwards). The justification of the hydrography knowledge gap with only NOAA internet fact sheets is also weak: there have been many technical reports and white papers by bigger players (e.g. GEBCO) looking at the rest of the world, and in particular further offshore.

The gap between the work presented here and how it can be used in sonar imaging needs to be presented numerically: how do these results of a tank experiment, with ideal conditions, titanium oxide and a target made of metallic spheres imaged from tens of centimeters away scale up to actual field sonar imaging? How can other users be inspired to look at this interesting approach for their own applications? We look to Nature papers to influence thinking in the field, and this needs to be clear in any future revision.

This is interesting work, but how it is communicated and where are important factors to consider, to give it the audience and impact it deserves.

Reviewer #2 (Remarks to the Author):

The authors introduce a photoacoustic airborne sonar imaging concept, which can be used to obtain the image of an underwater target in hydrodynamic conditions. Along with the basic concept, the authors show the readers an experimental system to implement the photoacoustic imaging. The basic idea is very interesting and worth to be explored.

My biggest concern is that the transmitting transducer (served as an underwater acoustic source) in the experiment may not be able to replace the acoustic source generated by the photoacoustic effect of a strong laser beam impinging on the water surface. One of the most challenging problems is that, when the water surface (or, sea surface) is moving and fluctuating, whether the laser source could generate desired acoustic signals (at least, having enough source levels) for imaging processing. Although the authors have given some discussions on this problem, the readers will still wonder and guess the validity of the replacement of the laser-generated acoustic source with a real sonar transducer.

If possible, the direct use of a laser source will be more persuasive. If not, more discussions on this problem are required (may be, a section focusing on this topic will be more intuitive). Meanwhile, some necessary revisions on the abstract and conclusion parts are required.

Others suggestions.

1 transmitting hydrophone transmitting transducer.

2 Underwater target imaging is sometimes quite different from bathymetry. A target may be modeled as a group of several point-scatterers while the sea bed is always a continuous one. This difference will determine that the operating modes, the image-forming principles, the performance values of imaging sonar systems are different from each other. Hence, the authors are suggested to focus on one of the two topics, and revise the introduction part if necessary.

Reviewer #3 (Remarks to the Author):

The paper "Multi-modal sensor fusion for three-dimensional airborne sonar imaging in hydrodynamic conditions" introduces a very interesting concept on how to handle the transition from acoustic waves from dynamic water to the air in order to be acquired by an airborne platform. The underlying idea is very noteworthy and the algorithmic solution convinces with its simplicity. Actually, the possibility to really achieve an airborne sonar imaging would be great and the authors make a very big step in this direction.

However, the claims of the paper are too ambitious and are not yet justified by the given results. This also is the main shortcoming of the paper. Even though the fundamental idea of the paper is very interesting and the proposed multi-modal sensor fusion sounds reasonable, there are still too many open questions that would have to be examined before one can really speak of an airborne sonar imaging. Actually, the carried out experiments show that the authors may be on the right way, but there are too many assumptions that have been made to claim that a complete system is working as well in a practical scenario. To name just a few ones, adding TiO₂ to the water sounds very helpful for measuring the surface, but is practically impossible. Since an exact measurement of the water surface is of high importance, the problem on how to measure the surface without any tricks would be

required first, before one should speak of the first acoustic underwater imaging performed aerially. This holds especially since surface measurements of transparent material is an extremely challenging task. Furthermore, the scaling to a real practical usage is not properly discussed. The waves in open water are far more complex than the modeled ones and effects like currents, temperature gradients, sea spray and many more also would have to be discussed. Additionally, for an airborne measurement scenario, an access to the actual speed of sound in water would not be possible. However, errors in the estimation of the speed of sound may cause artifacts that also would have to be treated. Moreover, in the experiments the laser excitation was replaced by a hydrophone. Even though this is understandable for security reasons, without having tested the scenario with all components in the loop, it is not justified to really speak from an airborne system. Apparently, all this are only related problems and one might find solutions for most of them. However, before one can speak of a fully running airborne system, these challenges would also have to be addressed. Or respectively, the paper should focus only on its core contribution.

In this context, unfortunately the algorithm for the multi-modal sensor fusion is not properly described and would require more details. For example it is not discussed why an ROI is extracted from the depth map and for what this is required. Furthermore, a more detailed discussion on how the surface map can be used as a volumetric representation of the speed of sound is missing. Additionally, it is not discussed why only the distribution of the speed of sound is used. One might think for example why the surface angle is not used in addition. Apparently, not only the local height of the surface influences the reconstruction, but also the slope. Some of the information that helps the reader to understand the algorithm is given in the supplementary information. Actually, it may be more suited in the main paper as it is essential for understanding the paper.

Another point is that the description of the experiments is too superficial. This is especially true since the experiments only provide qualitative results and not quantifiable ones. For actually assessing the performance of the proposed concept, an evaluation of the different parameters influencing the reconstruction quality would be required. For example, it would be interesting to see how strong the influence of the accuracy of the depth measurements is. Or respectively how strongly the presence of interfering light sources like the sun (in an outdoor scenario) affects the surface estimation and therewith the reconstruction quality. Furthermore, the influence of the source SNR would be an important issue. Especially since the laser excitation has been replaced by a hydrophone, the influence of the excitation of the performance would have to be studied more closely. Also performance measurements for different wave heights would be very interesting to see.

Finally, the structure of paper is a bit uncommon and not beneficial for the readability. This holds especially for the order of the sections, as "Results" is given before "Methods" and a "Conclusion" is missing. Furthermore, the "Results" section is too long and should be split, especially since it contains the description of the reconstruction algorithm. Regarding the "Introduction", it may be advisable to split it in an actual introduction and an overview of the state of the art. Finally, the system information and the discussion of the setup is spread over several parts of the paper, therewith harming the overall comprehensibility.

Reviewer #4 (Remarks to the Author):

The authors proposed in the paper a novel sensor fusion method for airborne sonar imaging in hydrodynamic conditions. To compensate for hydrodynamic surface waves they used a depth camera to measure the surface condition and fused it with an acoustic signal captured from CMUT. The paper presents the comparison results and verified the proposed method. To the best of my knowledge, 3D airborne sonar imaging in hydrodynamic conditions is quite new. Even though some aspects need redefinition considering real dynamic ocean wave conditions, the approach and results are quite high quality.

The manuscript is clear, relevant for the field, and presented in a well-structured manner. The manuscript's results are reproducible based on the details given in the methods section. The figures/tables/images/schemes are appropriate. They properly show the data, they are easy to interpret and understand. The data are interpreted appropriately and consistently throughout the manuscript

MINOR ISSUE: The caption of Figure 5 needs to be corrected because Figure 5 presents a comparison of results in both hydrostatic and hydrodynamic conditions.

Key to responses:

- Reviewer's comments are in black (unedited).
- Authors' responses to the reviewers' comments are in blue.
- Quotations from the paper are in red and *italicized*.

Reviewer 1:

This manuscript proposes to demonstrate 3-D sonar imaging of underwater targets using an airborne system. After a rather long introduction to show that underwater sonars are not up to the task, there is a short description of the current state of photoacoustics. This is followed with a thorough description of the results and methodology, with the presentation of image reconstruction from tank experiments over an area of 12 by 16 cm (Figure 5), imaged from 60 cm above the water. These experiments are done with the addition of titanium dioxide in the water, and an embedded target constructed from metallic spheres, emplaced 18 cm deep. The addition of centimeter-high waves makes the task more complex, and the article nicely builds up on 2-D hydrostatic tests by the authors (reference 17).

Thank you for your feedback and acknowledgement of the complexity of our experimental work in comparison to our previous work. We have updated the introduction to be more concise, reducing the focus on conventional sonars and instead zeroing-in on the key challenges facing remote, airborne underwater imaging systems as well as on our core contributions.

The technical achievements are promising but I do not think this article is at the level expected of Nature Communications Engineering. I am therefore reluctantly arguing for major revisions, not because of the technical achievements but because of the way they are presented.

Thank you for the opportunity to demonstrate the impact of our work through a revision process. We have completely revamped the organization of our manuscript while refocusing and clarifying the core contributions of our work.

As we outline in the manuscript, the core contribution of this work is solving the fundamental challenge of imaging underwater environments in hydrodynamic conditions using airborne ultrasound through the presented multi-modal sensor fusion framework and GPW-SAR image reconstruction algorithm. While the non-planar, hydrodynamic water surface proves to be a major bottleneck in existing remote underwater imaging approaches, our airborne ultrasound detection leverages the fact the ultrasound is able to pass through the air-water interface – despite incurring non-negligible distortions. Our technique and framework involves mapping the surface of the water such that we can invert the distortion incurred as acoustic signals cross the non-planar air-water interface as part of the image reconstruction procedure.

Such a framework will be a fundamental piece of the greater challenge of developing end-to-end airborne sonar systems that can operate in real-world conditions. In addition to the developed framework, we also present new in-depth simulation-driven analysis of the required specifications of the surface mapping imager that will enable future deployment of our system and framework in realistic, uncontrolled scenarios.

This includes the absence of explanations of how any innovation would stand up in the application of sonar imaging. Sonar imaging typically covers areas several kilometres square, depths ranging from centimetres to kilometres, and objects that are much more challenging to detect than metallic spheres (less acoustic differences with their backgrounds). Water bodies do not have the benefit of added titanium oxide either. So, at the very least, I would have expected an explanation of how these lab experiments in a very constrained setting would apply to real-world applications.

While current demonstrations of the proposed photoacoustic airborne sonar system are relatively simple when compared to the maturity of conventional sonar imaging, it is important to note that our work aims to present proof-of-concept demonstrations of a fundamentally new technology that is being built from the ground-up. With that in mind, the work presented in this manuscript provides an important piece of the puzzle and will be instrumental in the development of future, scaled iterations of the system that will aim to achieve competitive performance metrics to conventional sonar systems while gaining the advantages of completely airborne operation. Another important note of our system is the fact that the in-water acoustic propagation is equivalent to that of conventional sonar and therefore decades of research and development can be leveraged to overcome many second-order challenges – rather than reinventing the wheel. With this in mind, we are focused on de-risking the differentiating challenges that will make-or-break such a system concept, for example, propagation through a hydrodynamic air-water interface.

Moreover, since the hydrodynamic air-water interface has negligible effect on the SNR of the received acoustic echoes, achievable imaging depth and/or target strength of various targets was not a primary focus of this paper. Instead, for interested readers, we have detailed descriptions and analysis of the link budget included in our previous work [32] as well as in Supplementary Note 1.

In addition, we have greatly added to the Discussion section of the manuscript to include simulation-driven analysis on scaling the presented framework and results to practical usage, i.e. in real-world conditions without addition of titanium dioxide to the water. More details on the new analysis included on scaling to practical usage are presented in the response to the reviewer's next comment.

[32] Fitzpatrick, A., Singhvi, A. & Arbabian, A. An airborne sonar system for underwater remote sensing and imaging. *IEEE Access* **8**, 189945–189959 (2020).

The authors are doing very good experimental work, and I remember enjoying their 2020 IEEE article. It nicely presented the use of capacitive micromachined ultrasonic transducers (CMUTs), it was backed up by a stronger and more relevant bibliography, and the general concept was better presented, e.g. in its Figure 1. Measurements of acoustic attenuation (e.g. Figure 7) and other constraints of airborne acquisition on acoustic imaging made these results very useful. The extension to 3-D and hydrodynamic conditions needs to do the same. The Supplementary Material provides relatively similar information, for example that some targets could be detected down to 100 m deep, when imaging at 50 kHz, with close to 18 frames per second, but this is not directly presented and not fully discussed.

Thank you for acknowledging the quality of our experimental work. Beginning with the introduction, we have greatly improved the quality and relevance of our bibliography as well as included two additional figures (*updated Figure 1 and updated Figure 2*) which clearly and intuitively illustrate the system concept.

Figure 1: Photoacoustic Airborne Sonar System. Schematic of proposed system with the laser excitation source, ultrasound detectors, and surface mapping imager all on-board an airborne platform which here is depicted as an Unmanned Aerial Vehicle (UAV).

Figure 2: Hydrostatic vs. Hydrodynamic Air-Water Interface. Top: Simulated forward propagation and image reconstruction in hydrostatic conditions. Middle: Simulated propagation in hydrodynamic conditions, but assumed hydrostatic in image reconstruction. Bottom: Simulated propagation in hydrodynamic conditions, with correct hydrodynamic channel in image reconstruction.

Additionally, we have greatly added to the Discussion section of the manuscript by including new simulation-driven analysis which provides context on the required surface mapping accuracy, spatial resolution, and frame rate to scale these lab-based experiments in a water tank to actual deployment in real-world conditions. These simulation results (*updated Figure 8*) assess the impact of different

parameters on the quality of the reconstructed 3D image: 1) the utilized acoustic frequency, 2) the height of the receivers in the air, 3) the depth of the target in the water, and 4) the wave height of the surface waves. Through this analysis, we vary the accuracy of the surface mapping to identify which parameters have the greatest influence.

Not only does this help to identify surface mapping requirements for specific system operating conditions, but it also provides a landscape of the system's degrees-of-freedom and can be used to understand the trade-off space for the design of the next-generation surface mapping solution. For example, if relatively low resolution images of the underwater environment are sufficient, one could reduce the operating acoustic frequency, which in-turn reduces the required surface mapping accuracy. Another example pertains to the wave height, if the system is operating in an environment with larger wave heights, then one could employ a surface mapping imager with less stringent accuracy requirements. While the trade-off space is complex, this analysis provides readers with intuition about the tuning knobs afforded by the proposed multi-physics, multi-sensor system, which proves to be very insightful and we thank the reviewer for this suggestion.

Figure 8: Surface Mapping Accuracy Simulations. *a* Example surface map used in the simulations. *b* YZ-Cut of the example simulation setup showing the acoustic source at the water surface with waves that have peak-to-peak amplitude A , underwater target at depth D , and ultrasound (US) transducers at height H . Simulated image degradation with surface map errors as a function of: *c* acoustic frequency, *d* receiver height, *e* target depth, and *f* wave amplitude.

These results could also have advantageously been presented in the Abstract, currently devoid of any numerical information.

We have re-written the abstract to be more focused on the challenge, core contributions, and impact of the presented work while keeping it accessible to a non-specialist audience in line with the journal's guidelines (<https://www.nature.com/commseng/submit/submission-guidelines#online-submission>). Due to the complex trade-off space alluded to above, we have elected not to include numerical information in the abstract while still including relevant, specific system details.

The current bibliography presents some interesting articles but does not achieve the breadth of the references in the IEEE 2020 article, and it was interesting to see no reference to some of the pioneering photoacoustic work of V. Humphrey et al. (from the 1980s onwards). The justification of the hydrography knowledge gap with only NOAA internet fact sheets is also weak: there have been many technical reports and white papers by bigger players (e.g. GEBCO) looking at the rest of the world, and in particular further offshore.

Thank you for bringing this issue to our attention. In the process of revamping our introduction, we have a much more comprehensive bibliography, including several additional references that will provide readers with context on a breadth of topics such as the motivation and different application spaces for remote underwater imaging, prior work including conventional sonar and current airborne approaches, as well as other topics where relevant.

The gap between the work presented here and how it can be used in sonar imaging needs to be presented numerically: how do these results of a tank experiment, with ideal conditions, titanium oxide and a target made of metallic spheres imaged from tens of centimeters away scale up to actual field sonar imaging? How can other users be inspired to look at this interesting approach for their own applications? We look to Nature papers to influence thinking in the field, and this needs to be clear in any future revision.

This is interesting work, but how it is communicated and where are important factors to consider, to give it the audience and impact it deserves

We have greatly improved the presentation of our technical achievements through 1) clarity on the scope of the work presented in the manuscript, 2) enhanced linearity of the presented theory, technical contributions, results, and simulation-driven analysis, and 3) added simulation results which articulate the required future work that will enable using the multi-modal sensor fusion framework in real-world applications, i.e. without adding titanium dioxide to the water. In addition, on the note of scalability to greater depths and detection of various targets, we include a Link Budget Analysis in the Supplementary Note 1; we do not believe that this analysis belongs in the main manuscript as the focus of this work is on the sensor fusion framework and image reconstruction algorithm that allow PASS to overcome hydrodynamic conditions.

With these improvements, we believe that the updated manuscript elucidates how the proposed approach overcomes the major bottleneck presented by hydrodynamic air-water interfaces in existing remote underwater imaging approaches while also including additional analysis useful for researchers working on the increasingly important problem of remote underwater sensing. The proposed multi-physics, multi-sensor framework is a key piece in the complex puzzle of designing and building the first high-fidelity 3D airborne photoacoustic sonars, which is why we believe this work delivers great impact and makes it worthy of publication in the journal.

Reviewer 2:

The authors introduce a photoacoustic airborne sonar imaging concept, which can be used to obtain the image of an underwater target in hydrodynamic conditions. Along with the basic concept, the authors show the readers an experimental system to implement the photoacoustic imaging. The basic idea is very interesting and worth to be explored.

Thank you for your review and for recognizing the importance of this work.

My biggest concern is that the transmitting transducer (served as an underwater acoustic source) in the experiment may not be able to replace the acoustic source generated by the photoacoustic effect of a strong laser beam impinging on the water surface. One of the most challenging problems is that, when the water surface (or, sea surface) is moving and fluctuating, whether the laser source could generate desired acoustic signals (at least, having enough source levels) for imaging processing. Although the authors have given some discussions on this problem, the readers will still wonder and guess the validity of the replacement of the laser-generated acoustic source with a real sonar transducer.

If possible, the direct use of a laser source will be more persuasive. If not, more discussions on this problem are required (may be, a section focusing on this topic will be more intuitive).

We agree that direct use of the laser source would be more persuasive for the hydrodynamic experiments; however, with the current solution for water surface mapping it is required to add TiO_2 to the water to increase the optical reflectivity from the water surface. Unfortunately, the laser cannot be used with the TiO_2 -dyed water. Moreover, the primary challenge in imaging underwater environments in hydrodynamic conditions is the handling of the acoustic distortions incurred as a result of the airborne detection whereas the aspects of the airborne laser excitation are well-understood by theory and have been de-risked by other researchers. Thus, the core contributions of this work are the multi-modal sensor fusion framework and GPW-SAR algorithm that enable inverting the distortion incurred as acoustic signals cross the non-planar air-water interface in hydrodynamic conditions.

Nevertheless, to convince readers that future demonstrations of the end-to-end photoacoustic airborne sonar system, i.e. one that uses the laser source, could employ this framework, we include a dedicated *“Transmitter vs. Laser-Generated Source”* subsection to the Discussion, per your suggestion. In this subsection, we articulate that the acoustic signal transmitted by the transmitter is nearly equivalent to that generated by the laser, both in modulation and in pressure level, with the transmitter source pressure level being ~ 3 Pascals and the laser-generated source being ~ 1 Pascal. It is important to note that neither experiment was SNR-constrained and that these pressure levels could have been equivalent, but unfortunately at the time of experimentation, we simply estimated the order of magnitude of the pressure levels. Being the same order of magnitude, these sources can be approximated as equivalent.

Additionally, as mentioned in your review, it is more important to understand the effect of the laser being incident on the hydrodynamic water surface. Again, the dynamics of the water surface can be ignored due to the microsecond duration of the laser excitation. To mitigate the concern of the laser's oblique incidence on the non-planar water surface, we leverage well-understood theory to provide analysis which

illustrates that primary effect is a shift in the directivity of the generated underwater acoustic source, with only negligible variation in source pressure level over the range of expected incidence angles. Since the source is nearly isotropic, the shift in the directivity has minimal effect on the underwater insonification. The below figure is included in the manuscript to illustrate this analysis:

Figure 7: Oblique Laser Incidence. Left: Normalized pressure along the depth-axis ($\psi = 0^\circ$) as a function of the laser incidence angle for laser beam radii of 1 mm and 1 cm. Right: Comparison of underwater acoustic directivity for normal versus oblique incidence.

Future work will focus on developing a surface mapping solution that fits the specifications outlined in the Discussion section (i.e. millimeter-scale accuracy, centimeter-scale spatial resolution, and 18+ FPS frame rate) without introducing an additive to the water. At this time, we will be able to demonstrate PASS in an end-to-end fashion.

Meanwhile, some necessary revisions on the abstract and conclusion parts are required.

Thank you for your comment. We have revised the abstract and conclusions within the paper to better align to the core contributions of our work.

Others suggestions.

1 transmitting hydrophone transmitting transducer.

Thank you for your suggestion - we have updated this accordingly.

2 Underwater target imaging is sometimes quite different from bathymetry. A target may be modeled as a group of several point-scatterers while the sea bed is always a continuous one. This difference will determine that the operating modes, the image-forming principles, the performance values of imaging sonar systems are different from each other. Hence, the authors are suggested to focus on one of the two topics, and revise the introduction part if necessary

Thank you for surfacing this concern. We agree that operating modes, image formation principles, and performance criteria can be quite different, and while we believe that the proposed system could have application to both target imaging and bathymetry (and that both of these applications would need to overcome the acoustic distortion caused by a hydrodynamic water surface), we have removed mentions to bathymetry to eliminate potential confusion. Instead, we focus our attention to climate related applications as well as applications including disaster response, biological surveys, archaeology, and wreckage searching.

Reviewer 3:

The paper “Multi-modal sensor fusion for three-dimensional airborne sonar imaging in hydrodynamic conditions” introduces a very interesting concept on how to handle the transition from acoustic waves from dynamic water to the air in order to be acquired by an airborne platform. The underlying idea is very noteworthy and the algorithmic solution convinces with its simplicity. Actually, the possibility to really achieve an airborne sonar imaging would be great and the authors make a very big step in this direction.

Thank you for acknowledging the impact of our work.

However, the claims of the paper are too ambitious and are not yet justified by the given results. This also is the main shortcoming of the paper. Even though the fundamental idea of the paper is very interesting and the proposed multi-modal sensor fusion sounds reasonable, there are still too many open questions that would have to be examined before one can really speak of an airborne sonar imaging. Actually, the carried out experiments show that the authors may be on the right way, but there are too many assumptions that have been made to claim that a complete system is working as well in a practical scenario. To name just a few ones, adding TiO_2 to the water sounds very helpful for measuring the surface, but is practically impossible. Since an exact measurement of the water surface is of high importance, the problem on how to measure the surface without any tricks would be required first, before one should speak of the first acoustic underwater imaging performed aurally. This holds especially since surface measurements of transparent material is an extremely challenging task.

Thank you for bringing this concern to our attention. After further review of our work in consideration of this feedback, we have refocused the claims made in the updated manuscript as described below. While our hydrostatic imaging results are in fact the first demonstration of “three-dimensional acoustic imaging of underwater from an airborne system,” and our hydrodynamic imaging results utilize fully airborne acoustic detectors, the hydrodynamic imaging results required the addition of TiO_2 to the water and as a result used the in-water acoustic transmitter, which has been clearly reflected in the updated manuscript. That said, we believe, as stated in your review, that this work is a major milestone in moving towards a fully airborne imaging system as it addresses the fundamental challenge of overcoming the acoustic distortions incurred as a result of the airborne detection.

Throughout the manuscript and based on the feedback, we have now put more emphasis on our core contributions: the presented multi-modal sensor fusion framework and GPW-SAR image reconstruction algorithm that enable inverting the distortion incurred as acoustic signals cross the non-planar air-water

interface in hydrodynamic conditions. In addition, we now dedicate the Discussion section of the manuscript entirely to the assumptions in the hydrodynamic experiments and analysis on the required specifications of a surface mapping imager that will enable our framework to be deployed in realistic, uncontrolled scenarios in the future. Through these means, we believe that the scope of our claims are now in-line with the impact of our contributions while leaving the readers aware of the future work.

Furthermore, the scaling to a real practical usage is not properly discussed. The waves in open water are far more complex than the modeled ones and effects like currents, temperature gradients, sea spray and many more also would have to be discussed. Additionally, for an airborne measurement scenario, an access to the actual speed of sound in water would not be possible. However, errors in the estimation of the speed of sound may cause artifacts that also would have to be treated. Moreover, in the experiments the laser excitation was replaced by a hydrophone. Even though this is understandable for security reasons, without having tested the scenario with all components in the loop, it is not justified to really speak from an airborne system. Apparently, all this are only related problems and one might find solutions for most of them. However, before one can speak of a fully running airborne system, these challenges would also have to be addressed. Or respectively, the paper should focus only on its core contribution.

Thank you for the detailed feedback. As mentioned above, we shifted the tone of the paper and scope of our claims to focus only on the core contributions: the presented multi-modal sensor fusion framework and GPW-SAR image reconstruction algorithm. Our goal with this manuscript is to answer the fundamental question: “with an accurate map of the water surface, can we reconstruct high-fidelity images of the underwater environment?” To answer this question, we emphasize our work on the framework, algorithms, and analysis of surface mapping requirements that make this possible.

That said, we do want the readers to understand how this framework could someday be used by the end-to-end photoacoustic airborne sonar system. Therefore, in the Discussion section, we added a subsection that explicitly focuses on the substitution of the laser source for the underwater acoustic transmitter. Here, we explain 1) that the source modulations are equivalent, 2) that the source pressure levels are nearly equivalent (~ 3 Pascals for the transmitter and ~ 1 Pascal for the laser-generated source), and 3) that there is negligible impact of the non-planar water surface on the incident laser. Through these three factors, we conclude the validity of the source substitution.

In addition, we updated the Discussion section to include simulation-driven analysis on scaling the presented results to practical usage. This section analyzes the required surface mapping accuracy, spatial resolution, and frame rate that will enable using the presented framework and image reconstruction algorithm in practical deployment of the photoacoustic airborne sonar system. While we agree that second-order complexities of water waves including currents, sea spray, whitecaps, large swells, etc. will need to be analyzed further, we believe that this is out of the scope of the current work, and thus, we have included a note on second-order complexities to the summary of our future work in the manuscript:

“Future work will also analyze second-order wave effects, such as sea spray, whitecaps, large swells, etc. that may require additional mitigation strategies.”

Another notable aspect mentioned in your review is the impact of unknown speed-of-sound in the water. If there is a mismatch between the true speed-of-sound and the speed-of-sound modeled in the reconstruction algorithm, image quality degradation is to be expected. That said, once our GPW-SAR algorithm compensates for the distortions across the air-water interface, the reconstruction in the water becomes equivalent to a classic in-water sonar reconstruction problem. This issue of unknown speed-of-sound as a function of water depth is well-studied and well-modeled in the sonar literature, and thus future iterations of the GPW-SAR algorithm can leverage decades of research and algorithms that help to solve this problem [S14-S16]. Consequently, it is most important that our modeled speed-of-sound has minimal error at the water surface so that the distortions can be appropriately compensated. We have added “*Supplementary Note 3*” to the supplemental material which provides detailed simulation-driven analysis of the impact of the unknown speed-of-sound in water on the image reconstruction quality. We did not believe that this analysis belonged in the main manuscript, though interested readers may find the analysis in the supplementary note useful.

- [S14] Wahl, D. E., Eichel, P., Ghiglia, D. & Jakowatz, C. Phase gradient autofocus—a robust tool for high resolution SAR phase correction. *IEEE Transactions on Aerospace and Electronic Systems* 30, 827–835 (1994).
- [S15] Piper, J. E. & Sternlicht, D. D. A low-order autofocus algorithm in OCEANS’11 MTS/IEEE KONA (2011), 1–4.
- [S16] Medwin, H. Speed of sound in water: A simple equation for realistic parameters. *The Journal of the Acoustical Society of America* 58, 1318–1319 (1975).

In this context, unfortunately the algorithm for the multi-modal sensor fusion is not properly described and would require more details. For example it is not discussed why an ROI is extracted from the depth map and for what this is required. Furthermore, a more detailed discussion on how the surface map can be used as a volumetric representation of the speed of sound is missing. Additionally, it is not discussed why only the distribution of the speed of sound is used. One might think for example why the surface angle is not used in addition. Apparently, not only the local height of the surface influences the reconstruction, but also the slope. Some of the information that helps the reader to understand the algorithm is given in the supplementary information. Actually, it may be more suited in the main paper as it is essential for understanding the paper.

Thank you for bringing to our attention the lack of clarity on some aspects of the presented algorithm. We have added significantly more detail on how and why we translate the acquired surface map to a volumetric representation of the speed of sound, in addition to altering the figure to better illustrate this.

“the surface map is converted into a discretized 3D volumetric representation of the acoustic channel defined over space, $c(x,y,z)$, where voxels above the water surface are assigned the speed-of-sound in air, c_{air} , and voxels beneath the water surface are assigned the speed-of-sound in water, c_{water} . With this model of the propagation channel, along with the corresponding acoustic measurements, an image reconstruction algorithm can now migrate the signals through the water surface while compensating for the distortions... [In] homogeneous media... a global constant value of [the speed-of-sound] can be assumed everywhere in space... On the other hand, imaging in heterogeneous media, for example across the air-water boundary, requires an accurate understanding of the speed-of-sound as a function of space,

i.e. $c(x,y,z)$. Above, we referred to $c(x,y,z)$ as the channel model, as it fully encapsulates the required information to understand the relationship between the temporal acoustic measurements captured by the ultrasonic transducers and the unknown target that we desire to reconstruct.”

Figure 3: Multi-Modal Sensor Fusion. Ultrasound transducers capture acoustic signals while a 3D imager maps the surface of the water. The raw sensor data are pre-processed to generate the complementary acoustic measurements and channel model which are consumed by an image reconstruction algorithm.

As you also mention in your feedback, we do not need to explicitly account for the surface angle (i.e. refraction) in our reconstruction algorithm. This is a major advantage of performing the reconstruction in the spectral domain and the following has been added to the manuscript to reflect that:

“An interesting note is that it is not required to explicitly account for refraction as we migrate the signals through the air-water interface in Steps 4-5. This is another advantage of using a spectral propagator in the spectral-frequency domain rather than a spatial propagator in the space-time domain as refraction is inherently handled by the transition of dispersion relation (i.e. k_z^a vs. k_z^w) as we cross the air-water interface.”

Lastly, an ROI is extracted from the depth map to ensure that the same spatial extent of the water surface is being considered for each of the measurements along the raster scan. Since the depth sensor is colocated with the ultrasound transducer, the field-of-view of the depth sensor varies as we raster scan the transducer. This is simply an implementation detail of our experimental setup and is not fundamental to the proposed framework and algorithm. As a result, we now present the framework in a generalized fashion and move this detail to the “Methods” section. In practical deployment, an ultrasound transducer array would be used and only a single surface map would need to be acquired. In this case, the system could utilize the full field-of-view of the surface mapping imager, rather than a smaller ROI. The following was added to the manuscript to provide clarity on this point:

“... we use a coded light depth sensor (Intel Realsense SR305) to capture a map of the water surface at each measurement location. The depth sensor is aligned adjacent to the CMUT with a known fixed offset. For each location, the surface map over a 26 cm x 24 cm region-of-interest (ROI) in the water tank is extracted, ensuring that the ROI is consistent across every measurement... It should be noted that in practical deployment, an ultrasound transducer array would be utilized such that only a single surface

map would need to be captured for the array of ultrasonic measurements. In this case, the full field-of-view of the surface mapping imager (rather than a smaller ROI) could be utilized and the GPW-SAR algorithm could be applied directly.”

Another point is that the description of the experiments is too superficial. This is especially true since the experiments only provide qualitative results and not quantifiable ones. For actually assessing the performance of the proposed concept, an evaluation of the different parameters influencing the reconstruction quality would be required. For example, it would be interesting to see how strong the influence of the accuracy of the depth measurements is. Or respectively how strongly the presence of interfering light sources like the sun (in an outdoor scenario) affects the surface estimation and therewith the reconstruction quality. Furthermore, the influence of the source SNR would be an important issue. Especially since the laser excitation has been replaced by a hydrophone, the influence of the excitation of the performance would have to be studied more closely. Also performance measurements for different wave heights would be very interesting to see.

Thank you for your feedback. To address these concerns, we have added in-depth simulation-driven analysis (*updated Figure 8*) to the Discussion section of the paper which assesses the different parameters influencing the reconstruction quality including 1) the utilized acoustic frequency, 2) the height of the receivers in the air, 3) the depth of the target in the water, and 4) the wave height of the surface waves. Through this analysis, we vary the accuracy of the surface mapping to identify which parameters have the greatest influence. Not only does this help to identify surface mapping requirements for specific system operating conditions, but it also provides a landscape of the system’s degrees-of-freedom and can be used to understand the trade-off space for the design of the next-generation surface mapping solution. For example, if relatively low resolution images of the underwater environment are sufficient, one could reduce the operating acoustic frequency, which in-turn reduces the required surface mapping accuracy. Another example pertains to the wave height, if the system is operating in an environment with larger wave heights, then one could employ a surface mapping imager with less stringent accuracy requirements. While the trade-off space is complex, this analysis provides readers with intuition about the tuning knobs afforded by the proposed multi-physics, multi-sensor system, which proves to be very insightful and we thank the reviewer for this suggestion.

While interfering light sources would prohibit the coded light depth sensor from being used in practical deployment, as noted in the manuscript, its use is simply for proof-of-concept demonstrations. That said, the next generation surface mapping solution that will be developed as part of future work will consider robustness to outdoor lighting conditions. We have added the below statement to the manuscript to reflect this consideration:

“... future work will involve developing an imager that is robust in outdoor lighting conditions and is capable of millimeter-scale accuracy, spatial resolution of ≤ 1 cm, and frame rate of ≥ 18 FPS -- without introducing an additive to the water.”

As mentioned in response to a previous comment, we have added the source pressure levels of the acoustic transmitter in comparison to the laser-generated source to provide transparency on the source SNR (~ 3 Pa vs. ~ 1 Pa). It is important to note that neither experiment was SNR-constrained and that these

pressure levels could have been equivalent, but unfortunately at the time of experimentation, we simply estimated the order of magnitude of the pressure levels. Being the same order of magnitude, these sources can be approximated as equivalent. Lastly, in Supplementary Note 1, we provide a link budget analysis on the imaging capabilities of the system (i.e. depth and target reflectivity) as a function of the laser-generated source strength.

Finally, the structure of paper is a bit uncommon and not beneficial for the readability. This holds especially for the order of the sections, as “Results” is given before “Methods” and a “Conclusion” is missing. Furthermore, the “Results” section is too long and should be split, especially since it contains the description of the reconstruction algorithm. Regarding the “Introduction”, it may be advisable to split it in an actual introduction and an overview of the state of the art. Finally, the system information and the discussion of the setup is spread over several parts of the paper, therewith harming the overall comprehensibility.

Thank you for this feedback. In our initial submission, we aimed to stay within the confines of the required sections and organization as dictated by the journal’s formatting guide:

(<https://www.nature.com/documents/commsj-phys-style-formatting-guide-accept.pdf>), i.e. Introduction, Results, Discussion, and Methods.

To improve the structure and organization of the manuscript, we have now included a “*Core Concepts*” section which provides a more logical flow of the presented ideas. This section incorporates delineation of hydrostatic vs. hydrodynamic conditions, an overview of the generalized multi-modal sensor fusion framework, and explanation of the image reconstruction algorithm. Including this section has greatly improved the readability of our manuscript and allows for the “Results” section to only cover the experimental 3D imaging results in hydrostatic and hydrodynamic conditions. In addition, we have implemented the experimental details provided in each of the “Results” and “Methods” sections as advised by the journal. Lastly, it is unconventional for the journal to include a “Related Works” and/or “Conclusion” sections, so we provide information about related works in the introduction and concluding information in the discussion section accordingly. Overall, we appreciate this feedback as it has greatly improved the linearity of our manuscript.

Reviewer 4:

The authors proposed in the paper a novel sensor fusion method for airborne sonar imaging in hydrodynamic conditions. To compensate for hydrodynamic surface waves they used a depth camera to measure the surface condition and fused it with an acoustic signal captured from CMUT. The paper presents the comparison results and verified the proposed method. To the best of my knowledge, 3D airborne sonar imaging in hydrodynamic conditions is quite new. Even though some aspects need redefinition considering real dynamic ocean wave conditions, the approach and results are quite high quality.

The manuscript is clear, relevant for the field, and presented in a well-structured manner.

The manuscript's results are reproducible based on the details given in the methods section.

The figures/tables/images/schemes are appropriate. They properly show the data, they are easy to interpret and understand. The data are interpreted appropriately and consistently throughout the manuscript

MINOR ISSUE: The caption of Figure 5 needs to be corrected because Figure 5 presents a comparison of results in both hydrostatic and hydrodynamic conditions.

Thank you for acknowledging the quality of our work and for bringing this overlooked issue to our attention. We have updated the caption of Figure 5 (*Figure 6* in the updated manuscript) accordingly.

Figure 6. Imaging in Hydrostatic and Hydrodynamic Conditions. *a* Schematic of the experimental setup where an acoustic transmitter replaces the laser excitation, a depth sensor (SR305) profiles the water surface, and the embedded target is 'U'-shaped. *b* Example point cloud captured by the SR305 and the corresponding channel model. *c* 3D reconstructed image in hydrostatic conditions. *d* 3D reconstructed image when surface waves are present but are not compensated. *e* 3D reconstructed image when surface waves are present and are properly modeled using the depth sensor. *f-h* Bird's-eye view of the reconstructed images in c-e.

REVIEWERS' COMMENTS:

Reviewer #1 (Remarks to the Author):

The core contribution of this work is to solve the fundamental challenge of imaging underwater environments using airborne ultrasound and multi-modal sensor fusion, with a proposed processing pipeline and image reconstruction algorithm. This article shows that ultrasound can transmit through the air-water interface, and how it is distorted. It also accounts for water surface movements. As such, it is therefore an interesting proof-of-concept.

The first version of this manuscript was reviewed by four reviewers, often with very similar comments but also with added points corresponding to their relative domains of expertise. The authors have constructively and fully engaged with these comments and incorporated them in the manuscript as and when necessary.

The revised Abstract, in line with the journal's guidelines, is much better at defining the exact scope of this article. Along with the revised text throughout the manuscript, it also better outlines inherent limitations, key achievements, and what will need to be future work.

Supplementary Note 1 offers welcome details of the PASS equation and Supplementary Figure 1 offers theoretical expectations of the maximum imaging depth. Apart from very large (whale) or very reflective (sphere) objects, this is mostly well below 50 m. This would match expectations based on experience in the field. The assumed frequency of 50 kHz is also an adequate trade-off, allowing to neglect the role of facet scattering from any waves at the sea surface (real bodies of water are rarely completely smooth). This is addressed in Supplementary Note 2.

I am therefore satisfied now that this fully revised article is suitable for publication.

Reviewer #2 (Remarks to the Author):

The authors have revised the manuscript carefully and thoroughly. Most importantly, they have added a whole section to discuss the reasonability of replacing the laser source (serving as an underwater acoustic source via the photo-acoustic effect) with a real underwater transmitter (directly converting the electrical energy into the sound energy). According to the explanation, the acoustic signal transmitted by the underwater transmitting transducer may have a similar modulation form and source level to that transmitted by the laser-generated photo-acoustic effect, which can remove my biggest doubt. I believe that the current version of this manuscript is acceptable.

Reviewer #3 (Remarks to the Author):

The authors of the paper entitled "Multi-modal sensor fusion towards three-dimensional airborne sonar imaging in hydrodynamic conditions" have done a good job of implementing the comments raised in the last review very well. The paper clearly benefits from the revision and the focus of the work is now very clear.

Key to responses:

- Reviewer's comments are in black (unedited).
- Authors' responses to the reviewers' comments are in blue.
- Quotations from the paper are in red and *italicized*.

Round 1:**Reviewer 1:**

This manuscript proposes to demonstrate 3-D sonar imaging of underwater targets using an airborne system. After a rather long introduction to show that underwater sonars are not up to the task, there is a short description of the current state of photoacoustics. This is followed with a thorough description of the results and methodology, with the presentation of image reconstruction from tank experiments over an area of 12 by 16 cm (Figure 5), imaged from 60 cm above the water. These experiments are done with the addition of titanium dioxide in the water, and an embedded target constructed from metallic spheres, emplaced 18 cm deep. The addition of centimeter-high waves makes the task more complex, and the article nicely builds up on 2-D hydrostatic tests by the authors (reference 17).

Thank you for your feedback and acknowledgement of the complexity of our experimental work in comparison to our previous work. We have updated the introduction to be more concise, reducing the focus on conventional sonars and instead zeroing-in on the key challenges facing remote, airborne underwater imaging systems as well as on our core contributions.

The technical achievements are promising but I do not think this article is at the level expected of Nature Communications Engineering. I am therefore reluctantly arguing for major revisions, not because of the technical achievements but because of the way they are presented.

Thank you for the opportunity to demonstrate the impact of our work through a revision process. We have completely revamped the organization of our manuscript while refocusing and clarifying the core contributions of our work.

As we outline in the manuscript, the core contribution of this work is solving the fundamental challenge of imaging underwater environments in hydrodynamic conditions using airborne ultrasound through the presented multi-modal sensor fusion framework and GPW-SAR image reconstruction algorithm. While the non-planar, hydrodynamic water surface proves to be a major bottleneck in existing remote underwater imaging approaches, our airborne ultrasound detection leverages the fact the ultrasound is able to pass through the air-water interface – despite incurring non-negligible distortions. Our technique and framework involves mapping the surface of the water such that we can invert the distortion incurred as acoustic signals cross the non-planar air-water interface as part of the image reconstruction procedure.

Such a framework will be a fundamental piece of the greater challenge of developing end-to-end airborne sonar systems that can operate in real-world conditions. In addition to the developed framework, we also present new in-depth simulation-driven analysis of the required specifications of the surface mapping

imager that will enable future deployment of our system and framework in realistic, uncontrolled scenarios.

This includes the absence of explanations of how any innovation would stand up in the application of sonar imaging. Sonar imaging typically covers areas several kilometres square, depths ranging from centimetres to kilometres, and objects that are much more challenging to detect than metallic spheres (less acoustic differences with their backgrounds). Water bodies do not have the benefit of added titanium oxide either. So, at the very least, I would have expected an explanation of how these lab experiments in a very constrained setting would apply to real-world applications.

While current demonstrations of the proposed photoacoustic airborne sonar system are relatively simple when compared to the maturity of conventional sonar imaging, it is important to note that our work aims to present proof-of-concept demonstrations of a fundamentally new technology that is being built from the ground-up. With that in mind, the work presented in this manuscript provides an important piece of the puzzle and will be instrumental in the development of future, scaled iterations of the system that will aim to achieve competitive performance metrics to conventional sonar systems while gaining the advantages of completely airborne operation. Another important note of our system is the fact that the in-water acoustic propagation is equivalent to that of conventional sonar and therefore decades of research and development can be leveraged to overcome many second-order challenges – rather than reinventing the wheel. With this in mind, we are focused on de-risking the differentiating challenges that will make-or-break such a system concept, for example, propagation through a hydrodynamic air-water interface.

Moreover, since the hydrodynamic air-water interface has negligible effect on the SNR of the received acoustic echoes, achievable imaging depth and/or target strength of various targets was not a primary focus of this paper. Instead, for interested readers, we have detailed descriptions and analysis of the link budget included in our previous work [32] as well as in Supplementary Note 1.

In addition, we have greatly added to the Discussion section of the manuscript to include simulation-driven analysis on scaling the presented framework and results to practical usage, i.e. in real-world conditions without addition of titanium dioxide to the water. More details on the new analysis included on scaling to practical usage are presented in the response to the reviewer's next comment.

[32] Fitzpatrick, A., Singhvi, A. & Arbabian, A. An airborne sonar system for underwater remote sensing and imaging. *IEEE Access* **8**, 189945–189959 (2020).

The authors are doing very good experimental work, and I remember enjoying their 2020 IEEE article. It nicely presented the use of capacitive micromachined ultrasonic transducers (CMUTs), it was backed up by a stronger and more relevant bibliography, and the general concept was better presented, e.g. in its Figure 1. Measurements of acoustic attenuation (e.g. Figure 7) and other constraints of airborne acquisition on acoustic imaging made these results very useful. The extension to 3-D and hydrodynamic conditions needs to do the same. The Supplementary Material provides relatively similar information, for example that some targets could be detected down to 100 m deep, when imaging at 50 kHz, with close to 18 frames per second, but this is not directly presented and not fully discussed.

Thank you for acknowledging the quality of our experimental work. Beginning with the introduction, we have greatly improved the quality and relevance of our bibliography as well as included two additional figures (*updated Figure 1 and updated Figure 2*) which clearly and intuitively illustrate the system concept.

Figure 1: Photoacoustic Airborne Sonar System. Schematic of proposed system with the laser excitation source, ultrasound detectors, and surface mapping imager all on-board an airborne platform which here is depicted as an Unmanned Aerial Vehicle (UAV).

Figure 2: Hydrostatic vs. Hydrodynamic Air-Water Interface. Top: Simulated forward propagation and image reconstruction in hydrostatic conditions. Middle: Simulated propagation in hydrodynamic conditions, but assumed hydrostatic in image reconstruction. Bottom: Simulated propagation in hydrodynamic conditions, with correct hydrodynamic channel in image reconstruction.

Additionally, we have greatly added to the Discussion section of the manuscript by including new simulation-driven analysis which provides context on the required surface mapping accuracy, spatial resolution, and frame rate to scale these lab-based experiments in a water tank to actual deployment in

real-world conditions. These simulation results (*updated Figure 8*) assess the impact of different parameters on the quality of the reconstructed 3D image: 1) the utilized acoustic frequency, 2) the height of the receivers in the air, 3) the depth of the target in the water, and 4) the wave height of the surface waves. Through this analysis, we vary the accuracy of the surface mapping to identify which parameters have the greatest influence.

Not only does this help to identify surface mapping requirements for specific system operating conditions, but it also provides a landscape of the system's degrees-of-freedom and can be used to understand the trade-off space for the design of the next-generation surface mapping solution. For example, if relatively low resolution images of the underwater environment are sufficient, one could reduce the operating acoustic frequency, which in-turn reduces the required surface mapping accuracy. Another example pertains to the wave height, if the system is operating in an environment with larger wave heights, then one could employ a surface mapping imager with less stringent accuracy requirements. While the trade-off space is complex, this analysis provides readers with intuition about the tuning knobs afforded by the proposed multi-physics, multi-sensor system, which proves to be very insightful and we thank the reviewer for this suggestion.

Figure 8: Surface Mapping Accuracy Simulations. *a* Example surface map used in the simulations. *b* YZ-Cut of the example simulation setup showing the acoustic source at the water surface with waves that have peak-to-peak amplitude A , underwater target at depth D , and ultrasound (US) transducers at height H . Simulated image degradation with surface map errors as a function of: *c* acoustic frequency, *d* receiver height, *e* target depth, and *f* wave amplitude.

These results could also have advantageously been presented in the Abstract, currently devoid of any numerical information.

We have re-written the abstract to be more focused on the challenge, core contributions, and impact of the presented work while keeping it accessible to a non-specialist audience in line with the journal's guidelines (<https://www.nature.com/commseng/submit/submission-guidelines#online-submission>). Due to the complex trade-off space alluded to above, we have elected not to include numerical information in the abstract while still including relevant, specific system details.

The current bibliography presents some interesting articles but does not achieve the breadth of the references in the IEEE 2020 article, and it was interesting to see no reference to some of the pioneering photoacoustic work of V. Humphrey et al. (from the 1980s onwards). The justification of the hydrography knowledge gap with only NOAA internet fact sheets is also weak: there have been many technical reports and white papers by bigger players (e.g. GEBCO) looking at the rest of the world, and in particular further offshore.

Thank you for bringing this issue to our attention. In the process of revamping our introduction, we have a much more comprehensive bibliography, including several additional references that will provide readers with context on a breadth of topics such as the motivation and different application spaces for remote underwater imaging, prior work including conventional sonar and current airborne approaches, as well as other topics where relevant.

The gap between the work presented here and how it can be used in sonar imaging needs to be presented numerically: how do these results of a tank experiment, with ideal conditions, titanium oxide and a target made of metallic spheres imaged from tens of centimeters away scale up to actual field sonar imaging? How can other users be inspired to look at this interesting approach for their own applications? We look to Nature papers to influence thinking in the field, and this needs to be clear in any future revision.

This is interesting work, but how it is communicated and where are important factors to consider, to give it the audience and impact it deserves

We have greatly improved the presentation of our technical achievements through 1) clarity on the scope of the work presented in the manuscript, 2) enhanced linearity of the presented theory, technical contributions, results, and simulation-driven analysis, and 3) added simulation results which articulate the required future work that will enable using the multi-modal sensor fusion framework in real-world applications, i.e. without adding titanium dioxide to the water. In addition, on the note of scalability to greater depths and detection of various targets, we include a Link Budget Analysis in the Supplementary Note 1; we do not believe that this analysis belongs in the main manuscript as the focus of this work is on the sensor fusion framework and image reconstruction algorithm that allow PASS to overcome hydrodynamic conditions.

With these improvements, we believe that the updated manuscript elucidates how the proposed approach overcomes the major bottleneck presented by hydrodynamic air-water interfaces in existing remote underwater imaging approaches while also including additional analysis useful for researchers working on the increasingly important problem of remote underwater sensing. The proposed multi-physics, multi-sensor framework is a key piece in the complex puzzle of designing and building the first high-fidelity 3D airborne photoacoustic sonars, which is why we believe this work delivers great impact and makes it worthy of publication in the journal.

Reviewer 2:

The authors introduce a photoacoustic airborne sonar imaging concept, which can be used to obtain the image of an underwater target in hydrodynamic conditions. Along with the basic concept, the authors show the readers an experimental system to implement the photoacoustic imaging. The basic idea is very interesting and worth to be explored.

Thank you for your review and for recognizing the importance of this work.

My biggest concern is that the transmitting transducer (served as an underwater acoustic source) in the experiment may not be able to replace the acoustic source generated by the photoacoustic effect of a strong laser beam impinging on the water surface. One of the most challenging problems is that, when the water surface (or, sea surface) is moving and fluctuating, whether the laser source could generate desired acoustic signals (at least, having enough source levels) for imaging processing. Although the authors have given some discussions on this problem, the readers will still wonder and guess the validity of the replacement of the laser-generated acoustic source with a real sonar transducer.

If possible, the direct use of a laser source will be more persuasive. If not, more discussions on this problem are required (may be, a section focusing on this topic will be more intuitive).

We agree that direct use of the laser source would be more persuasive for the hydrodynamic experiments; however, with the current solution for water surface mapping it is required to add TiO_2 to the water to increase the optical reflectivity from the water surface. Unfortunately, the laser cannot be used with the TiO_2 -dyed water. Moreover, the primary challenge in imaging underwater environments in hydrodynamic conditions is the handling of the acoustic distortions incurred as a result of the airborne detection whereas the aspects of the airborne laser excitation are well-understood by theory and have been de-risked by other researchers. Thus, the core contributions of this work are the multi-modal sensor fusion framework and GPW-SAR algorithm that enable inverting the distortion incurred as acoustic signals cross the non-planar air-water interface in hydrodynamic conditions.

Nevertheless, to convince readers that future demonstrations of the end-to-end photoacoustic airborne sonar system, i.e. one that uses the laser source, could employ this framework, we include a dedicated *“Transmitter vs. Laser-Generated Source”* subsection to the Discussion, per your suggestion. In this subsection, we articulate that the acoustic signal transmitted by the transmitter is nearly equivalent to that generated by the laser, both in modulation and in pressure level, with the transmitter source pressure level being ~ 3 Pascals and the laser-generated source being ~ 1 Pascal. It is important to note that neither experiment was SNR-constrained and that these pressure levels could have been equivalent, but unfortunately at the time of experimentation, we simply estimated the order of magnitude of the pressure levels. Being the same order of magnitude, these sources can be approximated as equivalent.

Additionally, as mentioned in your review, it is more important to understand the effect of the laser being incident on the hydrodynamic water surface. Again, the dynamics of the water surface can be ignored due to the microsecond duration of the laser excitation. To mitigate the concern of the laser's oblique

incidence on the non-planar water surface, we leverage well-understood theory to provide analysis which illustrates that primary effect is a shift in the directivity of the generated underwater acoustic source, with only negligible variation in source pressure level over the range of expected incidence angles. Since the source is nearly isotropic, the shift in the directivity has minimal effect on the underwater insonification. The below figure is included in the manuscript to illustrate this analysis:

Figure 7: Oblique Laser Incidence. Left: Normalized pressure along the depth-axis ($\psi = 0^\circ$) as a function of the laser incidence angle for laser beam radii of 1 mm and 1 cm. Right: Comparison of underwater acoustic directivity for normal versus oblique incidence.

Future work will focus on developing a surface mapping solution that fits the specifications outlined in the Discussion section (i.e. millimeter-scale accuracy, centimeter-scale spatial resolution, and 18+ FPS frame rate) without introducing an additive to the water. At this time, we will be able to demonstrate PASS in an end-to-end fashion.

Meanwhile, some necessary revisions on the abstract and conclusion parts are required.

Thank you for your comment. We have revised the abstract and conclusions within the paper to better align to the core contributions of our work.

Others suggestions.

1 transmitting hydrophone transmitting transducer.

Thank you for your suggestion - we have updated this accordingly.

2 Underwater target imaging is sometimes quite different from bathymetry. A target may be modeled as a group of several point-scatterers while the sea bed is always a continuous one. This difference will determine that the operating modes, the image-forming principles, the performance values of imaging sonar systems are different from each other. Hence, the authors are suggested to focus on one of the two topics, and revise the introduction part if necessary

Thank you for surfacing this concern. We agree that operating modes, image formation principles, and performance criteria can be quite different, and while we believe that the proposed system could have application to both target imaging and bathymetry (and that both of these applications would need to overcome the acoustic distortion caused by a hydrodynamic water surface), we have removed mentions to bathymetry to eliminate potential confusion. Instead, we focus our attention to climate related applications as well as applications including disaster response, biological surveys, archaeology, and wreckage searching.

Reviewer 3:

The paper “Multi-modal sensor fusion for three-dimensional airborne sonar imaging in hydrodynamic conditions” introduces a very interesting concept on how to handle the transition from acoustic waves from dynamic water to the air in order to be acquired by an airborne platform. The underlying idea is very noteworthy and the algorithmic solution convinces with its simplicity. Actually, the possibility to really achieve an airborne sonar imaging would be great and the authors make a very big step in this direction.

Thank you for acknowledging the impact of our work.

However, the claims of the paper are too ambitious and are not yet justified by the given results. This also is the main shortcoming of the paper. Even though the fundamental idea of the paper is very interesting and the proposed multi-modal sensor fusion sounds reasonable, there are still too many open questions that would have to be examined before one can really speak of an airborne sonar imaging. Actually, the carried out experiments show that the authors may be on the right way, but there are too many assumptions that have been made to claim that a complete system is working as well in a practical scenario. To name just a few ones, adding TiO_2 to the water sounds very helpful for measuring the surface, but is practically impossible. Since an exact measurement of the water surface is of high importance, the problem on how to measure the surface without any tricks would be required first, before one should speak of the first acoustic underwater imaging performed aurally. This holds especially since surface measurements of transparent material is an extremely challenging task.

Thank you for bringing this concern to our attention. After further review of our work in consideration of this feedback, we have refocused the claims made in the updated manuscript as described below. While our hydrostatic imaging results are in fact the first demonstration of “three-dimensional acoustic imaging of underwater from an airborne system,” and our hydrodynamic imaging results utilize fully airborne acoustic detectors, the hydrodynamic imaging results required the addition of TiO_2 to the water and as a result used the in-water acoustic transmitter, which has been clearly reflected in the updated manuscript. That said, we believe, as stated in your review, that this work is a major milestone in moving towards a fully airborne imaging system as it addresses the fundamental challenge of overcoming the acoustic distortions incurred as a result of the airborne detection.

Throughout the manuscript and based on the feedback, we have now put more emphasis on our core contributions: the presented multi-modal sensor fusion framework and GPW-SAR image reconstruction algorithm that enable inverting the distortion incurred as acoustic signals cross the non-planar air-water

interface in hydrodynamic conditions. In addition, we now dedicate the Discussion section of the manuscript entirely to the assumptions in the hydrodynamic experiments and analysis on the required specifications of a surface mapping imager that will enable our framework to be deployed in realistic, uncontrolled scenarios in the future. Through these means, we believe that the scope of our claims are now in-line with the impact of our contributions while leaving the readers aware of the future work.

Furthermore, the scaling to a real practical usage is not properly discussed. The waves in open water are far more complex than the modeled ones and effects like currents, temperature gradients, sea spray and many more also would have to be discussed. Additionally, for an airborne measurement scenario, an access to the actual speed of sound in water would not be possible. However, errors in the estimation of the speed of sound may cause artifacts that also would have to be treated. Moreover, in the experiments the laser excitation was replaced by a hydrophone. Even though this is understandable for security reasons, without having tested the scenario with all components in the loop, it is not justified to really speak from an airborne system. Apparently, all this are only related problems and one might find solutions for most of them. However, before one can speak of a fully running airborne system, these challenges would also have to be addressed. Or respectively, the paper should focus only on its core contribution.

Thank you for the detailed feedback. As mentioned above, we shifted the tone of the paper and scope of our claims to focus only on the core contributions: the presented multi-modal sensor fusion framework and GPW-SAR image reconstruction algorithm. Our goal with this manuscript is to answer the fundamental question: “with an accurate map of the water surface, can we reconstruct high-fidelity images of the underwater environment?” To answer this question, we emphasize our work on the framework, algorithms, and analysis of surface mapping requirements that make this possible.

That said, we do want the readers to understand how this framework could someday be used by the end-to-end photoacoustic airborne sonar system. Therefore, in the Discussion section, we added a subsection that explicitly focuses on the substitution of the laser source for the underwater acoustic transmitter. Here, we explain 1) that the source modulations are equivalent, 2) that the source pressure levels are nearly equivalent (~ 3 Pascals for the transmitter and ~ 1 Pascal for the laser-generated source), and 3) that there is negligible impact of the non-planar water surface on the incident laser. Through these three factors, we conclude the validity of the source substitution.

In addition, we updated the Discussion section to include simulation-driven analysis on scaling the presented results to practical usage. This section analyzes the required surface mapping accuracy, spatial resolution, and frame rate that will enable using the presented framework and image reconstruction algorithm in practical deployment of the photoacoustic airborne sonar system. While we agree that second-order complexities of water waves including currents, sea spray, whitecaps, large swells, etc. will need to be analyzed further, we believe that this is out of the scope of the current work, and thus, we have included a note on second-order complexities to the summary of our future work in the manuscript:

“Future work will also analyze second-order wave effects, such as sea spray, whitecaps, large swells, etc. that may require additional mitigation strategies.”

Another notable aspect mentioned in your review is the impact of unknown speed-of-sound in the water. If there is a mismatch between the true speed-of-sound and the speed-of-sound modeled in the reconstruction algorithm, image quality degradation is to be expected. That said, once our GPW-SAR algorithm compensates for the distortions across the air-water interface, the reconstruction in the water becomes equivalent to a classic in-water sonar reconstruction problem. This issue of unknown speed-of-sound as a function of water depth is well-studied and well-modeled in the sonar literature, and thus future iterations of the GPW-SAR algorithm can leverage decades of research and algorithms that help to solve this problem [S14-S16]. Consequently, it is most important that our modeled speed-of-sound has minimal error at the water surface so that the distortions can be appropriately compensated. We have added “*Supplementary Note 3*” to the supplemental material which provides detailed simulation-driven analysis of the impact of the unknown speed-of-sound in water on the image reconstruction quality. We did not believe that this analysis belonged in the main manuscript, though interested readers may find the analysis in the supplementary note useful.

- [S14] Wahl, D. E., Eichel, P., Ghiglia, D. & Jakowatz, C. Phase gradient autofocus—a robust tool for high resolution SAR phase correction. *IEEE Transactions on Aerospace and Electronic Systems* 30, 827–835 (1994).
- [S15] Piper, J. E. & Sternlicht, D. D. A low-order autofocus algorithm in OCEANS’11 MTS/IEEE KONA (2011), 1–4.
- [S16] Medwin, H. Speed of sound in water: A simple equation for realistic parameters. *The Journal of the Acoustical Society of America* 58, 1318–1319 (1975).

In this context, unfortunately the algorithm for the multi-modal sensor fusion is not properly described and would require more details. For example it is not discussed why an ROI is extracted from the depth map and for what this is required. Furthermore, a more detailed discussion on how the surface map can be used as a volumetric representation of the speed of sound is missing. Additionally, it is not discussed why only the distribution of the speed of sound is used. One might think for example why the surface angle is not used in addition. Apparently, not only the local height of the surface influences the reconstruction, but also the slope. Some of the information that helps the reader to understand the algorithm is given in the supplementary information. Actually, it may be more suited in the main paper as it is essential for understanding the paper.

Thank you for bringing to our attention the lack of clarity on some aspects of the presented algorithm. We have added significantly more detail on how and why we translate the acquired surface map to a volumetric representation of the speed of sound, in addition to altering the figure to better illustrate this.

“the surface map is converted into a discretized 3D volumetric representation of the acoustic channel defined over space, $c(x,y,z)$, where voxels above the water surface are assigned the speed-of-sound in air, c_{air} , and voxels beneath the water surface are assigned the speed-of-sound in water, c_{water} . With this model of the propagation channel, along with the corresponding acoustic measurements, an image reconstruction algorithm can now migrate the signals through the water surface while compensating for the distortions... [In] homogeneous media... a global constant value of [the speed-of-sound] can be assumed everywhere in space... On the other hand, imaging in heterogeneous media, for example across the air-water boundary, requires an accurate understanding of the speed-of-sound as a function of space,

i.e. $c(x,y,z)$. Above, we referred to $c(x,y,z)$ as the channel model, as it fully encapsulates the required information to understand the relationship between the temporal acoustic measurements captured by the ultrasonic transducers and the unknown target that we desire to reconstruct.”

Figure 3: Multi-Modal Sensor Fusion. Ultrasound transducers capture acoustic signals while a 3D imager maps the surface of the water. The raw sensor data are pre-processed to generate the complementary acoustic measurements and channel model which are consumed by an image reconstruction algorithm.

As you also mention in your feedback, we do not need to explicitly account for the surface angle (i.e. refraction) in our reconstruction algorithm. This is a major advantage of performing the reconstruction in the spectral domain and the following has been added to the manuscript to reflect that:

“An interesting note is that it is not required to explicitly account for refraction as we migrate the signals through the air-water interface in Steps 4-5. This is another advantage of using a spectral propagator in the spectral-frequency domain rather than a spatial propagator in the space-time domain as refraction is inherently handled by the transition of dispersion relation (i.e. k_z^a vs. k_z^w) as we cross the air-water interface.”

Lastly, an ROI is extracted from the depth map to ensure that the same spatial extent of the water surface is being considered for each of the measurements along the raster scan. Since the depth sensor is colocated with the ultrasound transducer, the field-of-view of the depth sensor varies as we raster scan the transducer. This is simply an implementation detail of our experimental setup and is not fundamental to the proposed framework and algorithm. As a result, we now present the framework in a generalized fashion and move this detail to the “Methods” section. In practical deployment, an ultrasound transducer array would be used and only a single surface map would need to be acquired. In this case, the system could utilize the full field-of-view of the surface mapping imager, rather than a smaller ROI. The following was added to the manuscript to provide clarity on this point:

“... we use a coded light depth sensor (Intel Realsense SR305) to capture a map of the water surface at each measurement location. The depth sensor is aligned adjacent to the CMUT with a known fixed offset. For each location, the surface map over a 26 cm x 24 cm region-of-interest (ROI) in the water tank is extracted, ensuring that the ROI is consistent across every measurement... It should be noted that in practical deployment, an ultrasound transducer array would be utilized such that only a single surface

map would need to be captured for the array of ultrasonic measurements. In this case, the full field-of-view of the surface mapping imager (rather than a smaller ROI) could be utilized and the GPW-SAR algorithm could be applied directly.”

Another point is that the description of the experiments is too superficial. This is especially true since the experiments only provide qualitative results and not quantifiable ones. For actually assessing the performance of the proposed concept, an evaluation of the different parameters influencing the reconstruction quality would be required. For example, it would be interesting to see how strong the influence of the accuracy of the depth measurements is. Or respectively how strongly the presence of interfering light sources like the sun (in an outdoor scenario) affects the surface estimation and therewith the reconstruction quality. Furthermore, the influence of the source SNR would be an important issue. Especially since the laser excitation has been replaced by a hydrophone, the influence of the excitation of the performance would have to be studied more closely. Also performance measurements for different wave heights would be very interesting to see.

Thank you for your feedback. To address these concerns, we have added in-depth simulation-driven analysis (*updated Figure 8*) to the Discussion section of the paper which assesses the different parameters influencing the reconstruction quality including 1) the utilized acoustic frequency, 2) the height of the receivers in the air, 3) the depth of the target in the water, and 4) the wave height of the surface waves. Through this analysis, we vary the accuracy of the surface mapping to identify which parameters have the greatest influence. Not only does this help to identify surface mapping requirements for specific system operating conditions, but it also provides a landscape of the system’s degrees-of-freedom and can be used to understand the trade-off space for the design of the next-generation surface mapping solution. For example, if relatively low resolution images of the underwater environment are sufficient, one could reduce the operating acoustic frequency, which in-turn reduces the required surface mapping accuracy. Another example pertains to the wave height, if the system is operating in an environment with larger wave heights, then one could employ a surface mapping imager with less stringent accuracy requirements. While the trade-off space is complex, this analysis provides readers with intuition about the tuning knobs afforded by the proposed multi-physics, multi-sensor system, which proves to be very insightful and we thank the reviewer for this suggestion.

While interfering light sources would prohibit the coded light depth sensor from being used in practical deployment, as noted in the manuscript, its use is simply for proof-of-concept demonstrations. That said, the next generation surface mapping solution that will be developed as part of future work will consider robustness to outdoor lighting conditions. We have added the below statement to the manuscript to reflect this consideration:

“... future work will involve developing an imager that is robust in outdoor lighting conditions and is capable of millimeter-scale accuracy, spatial resolution of ≤ 1 cm, and frame rate of ≥ 18 FPS -- without introducing an additive to the water.”

As mentioned in response to a previous comment, we have added the source pressure levels of the acoustic transmitter in comparison to the laser-generated source to provide transparency on the source SNR (~ 3 Pa vs. ~ 1 Pa). It is important to note that neither experiment was SNR-constrained and that these

pressure levels could have been equivalent, but unfortunately at the time of experimentation, we simply estimated the order of magnitude of the pressure levels. Being the same order of magnitude, these sources can be approximated as equivalent. Lastly, in Supplementary Note 1, we provide a link budget analysis on the imaging capabilities of the system (i.e. depth and target reflectivity) as a function of the laser-generated source strength.

Finally, the structure of paper is a bit uncommon and not beneficial for the readability. This holds especially for the order of the sections, as “Results” is given before “Methods” and a “Conclusion” is missing. Furthermore, the “Results” section is too long and should be split, especially since it contains the description of the reconstruction algorithm. Regarding the “Introduction”, it may be advisable to split it in an actual introduction and an overview of the state of the art. Finally, the system information and the discussion of the setup is spread over several parts of the paper, therewith harming the overall comprehensibility.

Thank you for this feedback. In our initial submission, we aimed to stay within the confines of the required sections and organization as dictated by the journal’s formatting guide:

(<https://www.nature.com/documents/commsj-phys-style-formatting-guide-accept.pdf>), i.e. Introduction, Results, Discussion, and Methods.

To improve the structure and organization of the manuscript, we have now included a “*Core Concepts*” section which provides a more logical flow of the presented ideas. This section incorporates delineation of hydrostatic vs. hydrodynamic conditions, an overview of the generalized multi-modal sensor fusion framework, and explanation of the image reconstruction algorithm. Including this section has greatly improved the readability of our manuscript and allows for the “Results” section to only cover the experimental 3D imaging results in hydrostatic and hydrodynamic conditions. In addition, we have implemented the experimental details provided in each of the “Results” and “Methods” sections as advised by the journal. Lastly, it is unconventional for the journal to include a “Related Works” and/or “Conclusion” sections, so we provide information about related works in the introduction and concluding information in the discussion section accordingly. Overall, we appreciate this feedback as it has greatly improved the linearity of our manuscript.

Reviewer 4:

The authors proposed in the paper a novel sensor fusion method for airborne sonar imaging in hydrodynamic conditions. To compensate for hydrodynamic surface waves they used a depth camera to measure the surface condition and fused it with an acoustic signal captured from CMUT. The paper presents the comparison results and verified the proposed method. To the best of my knowledge, 3D airborne sonar imaging in hydrodynamic conditions is quite new. Even though some aspects need redefinition considering real dynamic ocean wave conditions, the approach and results are quite high quality.

The manuscript is clear, relevant for the field, and presented in a well-structured manner.

The manuscript's results are reproducible based on the details given in the methods section.

The figures/tables/images/schemes are appropriate. They properly show the data, they are easy to interpret and understand. The data are interpreted appropriately and consistently throughout the manuscript

MINOR ISSUE: The caption of Figure 5 needs to be corrected because Figure 5 presents a comparison of results in both hydrostatic and hydrodynamic conditions.

Thank you for acknowledging the quality of our work and for bringing this overlooked issue to our attention. We have updated the caption of Figure 5 (*Figure 6* in the updated manuscript) accordingly.

Figure 6. Imaging in Hydrostatic and Hydrodynamic Conditions. *a* Schematic of the experimental setup where an acoustic transmitter replaces the laser excitation, a depth sensor (SR305) profiles the water surface, and the embedded target is 'U'-shaped. *b* Example point cloud captured by the SR305 and the corresponding channel model. *c* 3D reconstructed image in hydrostatic conditions. *d* 3D reconstructed image when surface waves are present but are not compensated. *e* 3D reconstructed image when surface waves are present and are properly modeled using the depth sensor. *f-h* Bird's-eye view of the reconstructed images in *c-e*.

Round 2:

Reviewer 1:

The core contribution of this work is to solve the fundamental challenge of imaging underwater environments using airborne ultrasound and multi-modal sensor fusion, with a proposed processing pipeline and image reconstruction algorithm. This article shows that ultrasound can transmit through the air-water interface, and how it is distorted. It also accounts for water surface movements. As such, it is therefore an interesting proof-of-concept.

The first version of this manuscript was reviewed by four reviewers, often with very similar comments but also with added points corresponding to their relative domains of expertise. The authors have constructively and fully engaged with these comments and incorporated them in the manuscript as and when necessary.

The revised Abstract, in line with the journal's guidelines, is much better at defining the exact scope of this article. Along with the revised text throughout the manuscript, it also better outlines inherent limitations, key achievements, and what will need to be future work.

Supplementary Note 1 offers welcome details of the PASS equation and Supplementary Figure 1 offers theoretical expectations of the maximum imaging depth. Apart from very large (whale) or very reflective (sphere) objects, this is mostly well below 50 m. This would match expectations based on experience in the field. The assumed frequency of 50 kHz is also an adequate trade-off, allowing to neglect the role of facet scattering from any waves at the sea surface (real bodies of water are rarely completely smooth). This is addressed in Supplementary Note 2.

I am therefore satisfied now that this fully revised article is suitable for publication.

Thank you for your review and recommendation to publish our revised manuscript.

Reviewer 2:

The authors have revised the manuscript carefully and thoroughly. Most importantly, they have added a whole section to discuss the reasonability of replacing the laser source (serving as an underwater acoustic source via the photo-acoustic effect) with a real underwater transmitter (directly converting the electrical energy into the sound energy). According to the explanation, the acoustic signal transmitted by the underwater transmitting transducer may have a similar modulation form and source level to that transmitted by the laser-generated photo-acoustic effect, which can remove my biggest doubt. I believe that the current version of this manuscript is acceptable.

Thank you for your review and feedback throughout the review process. We believe that the reviewer suggestions have allowed us to greatly improve our manuscript.

Reviewer 3:

The authors of the paper entitled "Multi-modal sensor fusion towards three-dimensional airborne sonar imaging in hydrodynamic conditions" have done a good job of implementing the comments raised in the last review very well. The paper clearly benefits from the revision and the focus of the work is now very clear.

Thank you for your review and acknowledgement of our thorough responses. We agree that the manuscript benefited greatly from the reviewer feedback.